# From Generalist to Specialist Representation

Yujia Zheng [* 1 2]   Fan Feng [* 3 4]   Yuke Li [5]   Shaoan Xie [1]   Kevin Murphy [6]   Kun Zhang [1 4]

## Abstract

Given a generalist model, learning a task-relevant specialist representation is fundamental for downstream applications. Identifiability, the asymptotic guarantee of recovering the ground-truth representation, is critical because it sets the ultimate limit of any model, even with infinite data and computation. We study this problem in a completely nonparametric setting, without relying on interventions, parametric forms, or structural constraints. We first prove that the structure between time steps and tasks is identifiable in a fully unsupervised manner, even when sequences lack strict temporal dependence and may exhibit disconnections, and task assignments can follow arbitrarily complex and interleaving structures. We then prove that, within each time step, the task-relevant latent representation can be disentangled from the irrelevant part under a simple sparsity regularization, without any additional information or parametric constraints. Together, these results establish a hierarchical foundation: task structure is identifiable across time steps, and task-relevant latent representations are identifiable within each step. To our knowledge, each result provides a first general nonparametric identifiability guarantee, and together they mark a step toward provably moving from generalist to specialist models.

## 1. Introduction

Learning latent representations from high-dimensional observations is central to enabling machines to understand and act in the world (Bengio et al., 2013; Schölkopf et al., 2021). World models, for instance, compress raw sensory input into low-dimensional features that capture dynamics (Ha & Schmidhuber, 2018). Rather than modeling the entire environment, task-relevant representations are desirable because they retain only the information required for the task, providing both efficiency and robustness (Tishby & Zaslavsky, 2015; Wong et al., 2025). For instance, in autonomous driving, planning depends on the positions and velocities of nearby vehicles and pedestrians, not on the color of the cars or billboards along the road.

Without identifiability, a learned representation cannot be guaranteed to reflect the ground truth, even with infinite data and computation. This challenge has long been central to latent representation learning, extending beyond task-relevant settings (Hyvärinen & Pajunen, 1999; Locatello et al., 2019). Given two observationally equivalent models $\mathbf{o} = f(\mathbf{s})$ and $\mathbf{o} = \hat{f}(\hat{\mathbf{s}})$, an arbitrary transformation $\phi$ may exist such that $\hat{\mathbf{s}} = \phi(\mathbf{s})$. In this case, the recovered latents need not correspond in any meaningful way to the true ones. Task-relevant variables, for example, may remain entangled with irrelevant factors, making it impossible to isolate what actually matters for the task. Such ambiguity introduces irreducible uncertainty into a machine's internal model of the world, constraining the ceiling of achievable intelligence and creating risks in high-stakes applications.

Existing theory provides conditions for identifiability of latent representations. In classical linear settings, identifiability can be obtained under additional parametric assumptions, for example in factor models with constraints on loadings (Anderson et al., 1956; Jöreskog, 1969; Shapiro, 1985), in linear Independent Component Analysis (ICA) via non-Gaussianity (Comon, 1994; Hyvärinen et al., 2001), and in tensor or multi-view models via Kruskal-type rank conditions (Kruskal, 1977; Sidiropoulos & Bro, 2000; Allman et al., 2009). More recently, nonlinear theory has advanced along two routes. In nonlinear ICA, one line leverages auxiliary information across domains or time (Hyvärinen & Morioka, 2016; Hyvärinen et al., 2019; Yao et al., 2021; Hälvä et al., 2021; Lachapelle et al., 2022), while another constrains the mixing class (Taleb & Jutten, 1999; Moran et al., 2021; Kivva et al., 2022; Zheng et al., 2022; Gresele et al., 2021; Buchholz et al., 2022). In causal representation learning, identifiability is often derived from interventional data (von Kügelgen et al., 2023; Jiang & Aragam, 2023; Jin & Syrgkanis, 2023; Zhang et al., 2024; Varici et al., 2025) or counterfactual views (von Kügelgen et al., 2021; Brehmer et al., 2022), which require some control over the data-generating process. Recent work considers the gen-

[1]CMU [2]UIUC [3]UCSD [4]MBZUAI [5]UMD [6]UBC. Correspondence to: Yujia Zheng <yujiazh@cmu.edu>.

*Proceedings of the 43$^{rd}$ International Conference on Machine Learning*, Seoul, South Korea. PMLR 306, 2026. Copyright 2026 by the author(s).

eral setting without extra information, with the assumptions that both latent and observed variables are Boolean vectors (Zhang et al., 2025). These conditions provide significant insights into recovering the underlying generative process, but may overly restrict the range of applicable scenarios.

At the same time, most existing theoretical results focus on full identifiability of the latent system: either recovering all latent variables component-wisely, or identifying them up to ancestors or neighborhoods. Yet such comprehensive recovery is often unnecessary. In many applications, tasks depend only on a subset of latent factors – for instance, in robotic manipulation, success hinges on object pose and gripper position, while lighting and textures are irrelevant. Shifting the goal from full-system identifiability to task-relevant identifiability enables weaker assumptions while still directly supporting planning, transfer, and generalization. Recent works have explored subspace factorization (von Kügelgen et al., 2021; Kong et al., 2022; Li et al., 2023; Liu et al., 2023), aiming to decompose latent factors into interpretable blocks. However, these approaches impose fixed structures, such as content–style separation, and are not designed to accommodate flexible task settings, where latent variables may correspond to tasks with unknown number, structure, and assignment, and where this uncertainty can further vary across time steps. Thus, the question remains:

*Is a task-relevant world representation identifiable in the general setting?*

**Contributions.** To answer this, we develop a theoretical framework for identifying task-relevant representations from the complex dynamics of the observational world. Our first result proves that task structure across time is identifiable in a fully general setting, without any parametric or structural assumptions (*Section 3*). We do not require strict temporal dependence: steps may be disconnected or even i.i.d., and thus we cannot leverage the temporal information. In addition, tasks may appear, disappear, and reappear in arbitrary order, allowing interleaving task-time structures. After identifying the tasks for each time step, we further ask which latent variables are relevant to those tasks, and provide the first nonparametric identifiability result for task-relevant latent representations without relying on interventions or functional constraints (*Section 4*). Specifically, we show that fine-tuning a pretrained model with a simple task-latent regularization provably disentangles task-relevant variables from irrelevant ones. Together, these results mark a step towards establishing principled pathways from generalist to specialist models that achieve both compression and fidelity.

## 2. Preliminaries

We assume an observed sequence $\{\mathbf{o}_t\}_{t=1}^T$ generated by latent states $\{\mathbf{s}_t\}_{t=1}^T$, with $\mathbf{o}_t \in \mathbb{R}^{d_o}$, $\mathbf{s}_t \in \mathbb{R}^{d_s}$, and actions

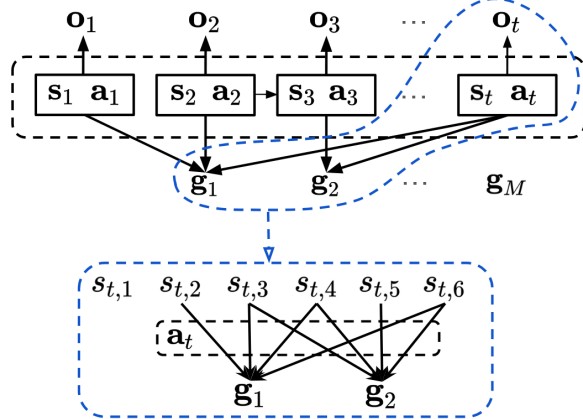

*Figure 1.* An illustration of the generative process. Latent states $\mathbf{s}_t$ generate observations $\mathbf{o}_t$ via nonlinear functions and interact with actions $\mathbf{a}_t$ under varying temporal connectivity, where consecutive steps may be arbitrarily disconnected. Tasks $\mathbf{g}_i$ are defined as colliders across time steps, and different tasks can arbitrarily interleave with one another. The zoomed-in view (right) shows how different components of $\mathbf{s}_t$ connect to multiple tasks via the intermediate actions.

$\mathbf{a}_t \in \mathbb{R}^{d_a}$. Observations satisfy

$$\mathbf{o}_t = f_t(\mathbf{s}_t), \qquad (1)$$

where $f_t$ is a diffeomorphism onto its image. The generative function $f_t$ is hidden and completely unknown. We allow varying temporal connectivity: $\mathbf{s}_t \to \mathbf{a}_t$ for all $t$, and $\mathbf{a}_t \to \mathbf{s}_{t+1}$, $\mathbf{s}_t \to \mathbf{s}_{t+1}$ whenever the boundary $t \to t+1$ is connected; both edges into $\mathbf{s}_{t+1}$ are omitted when it is disconnected. A Structural Causal Model (SCM) consistent with these is defined as $\mathbf{a}_t = \pi_t(\mathbf{s}_t, \eta_t)$, where

$$\mathbf{s}_{t+1} = \begin{cases} F_t(\mathbf{s}_t, \mathbf{a}_t, \xi_t), & \text{if } t \to t+1 \text{ is connected,} \\ F_t^0(\xi_t), & \text{otherwise,} \end{cases} \qquad (2)$$

with independent noises $\eta_t, \xi_t$. We define tasks $\{\mathbf{g}_i\}_{i=1}^M$ as colliders among different time steps, that is, $\mathbf{s}_t \to \mathbf{a}_t \to \mathbf{g}_i$ if the time step $t$ is relevant to $\mathbf{g}_i$. The visualization of the process is in Figure 1, and the reasons to define tasks as colliders instead of others are as follows:

**Remark 1** (**Why are tasks colliders?**)**.** *Modeling a shared task $\mathbf{g}_i$ as a collider is essential for capturing the coordinated nature of actions within a plan.*

- ***Confounder/Mediator:*** *The structures $\mathbf{a}_{t_1} \leftarrow \mathbf{g}_i \to \mathbf{a}_{t_2}$ or $\mathbf{a}_{t_1} \to \mathbf{g}_i \to \mathbf{a}_{t_2}$ would imply conditional independence: $\{\mathbf{s}_{t_1}, \mathbf{a}_{t_1}\} \perp\!\!\!\perp \{\mathbf{s}_{t_2}, \mathbf{a}_{t_2}\} \mid \mathbf{g}_i$. This is unrealistic as it treats steps within a task as isolated events rather than parts of a coherent strategy.*

- ***Collider:*** *The structure $\mathbf{a}_{t_1} \to \mathbf{g}_i \leftarrow \mathbf{a}_{t_2}$ correctly induces conditional dependence: $\{\mathbf{s}_{t_1}, \mathbf{a}_{t_1}\} \not\!\perp\!\!\!\perp \{\mathbf{s}_{t_2}, \mathbf{a}_{t_2}\} \mid \mathbf{g}_i$. This captures the intuition that time*

*steps within a task are interdependent, since they all target the same task.*

Given the observed variables $\{\mathbf{o}_t\}_{t=1}^{T}$ and the global set of tasks $\{\mathbf{g}_i\}_{i=1}^{M}$, our goal is first to identify the structure linking time steps and tasks (Section 3), and then, within each latent state $\mathbf{s}_t$, to isolate the components relevant to the associated tasks (Section 4). All theoretical guarantees need to be achieved in the general nonparametric setting without additional information.

## 3. Learning Temporal Task Structure

We first establish the identifiability of the time-task structure in the general setting. This structure is essential, as it forms the foundation for recovering task-relevant latent representations within each step. Without knowing which tasks are active at which times, disentangling latent variables at the step level would be ill-posed. Providing formal guarantee in the most general scenario is challenging, mainly due to the following reasons:

- The hidden process is fully nonparametric, with no auxiliary information or distributional constraints.

- Tasks may interleave arbitrarily over time, while classical decomposition assumes sequential completion.

- Temporal dependence is not guaranteed; the sequence may contain arbitrary disconnected boundaries.

Despite these challenges, we prove that the structure between time steps and tasks is identifiable under standard conditions. This result forms the first pillar of our framework: a principled characterization of temporal task structure in the general regime without additional information.

### 3.1. Characterization of Pair-wise Structure

We assume $T$ time steps, partitioned into $N$ contiguous segments of equal length $L = T/N$, with $L \geq 2$ and $N \mid T$. Let us define that

$$\mathbf{S} = \{\mathbf{S}_1, \ldots, \mathbf{S}_N\}, \qquad \mathbf{S}_i = \{\mathbf{s}_{(i-1)L+1}, \ldots, \mathbf{s}_{iL}\}. \quad (3)$$

All states within a segment share the same set of active tasks, and each task $\mathbf{g}_i$ must appear in at least two segments. Segments can be short (as few as two steps), ensuring flexibility in capturing state changes. To formalize the conditions used in our theory, we introduce the following notion.

**Definition 1** (Band conditioning set)**.** *For $k < v$ and task $\mathbf{g}_i$, define*

$$\mathbf{Z}_{\text{band}}(k, v, i) = \{\mathbf{s}_{kL-1}, \mathbf{s}_{kL+1}, \mathbf{s}_{vL-1}, \mathbf{s}_{vL+1}\}$$
$$\cap \{\mathbf{s}_1, \ldots, \mathbf{s}_T\} \cup \{\mathbf{g}_i\},$$

*with out-of-range indices omitted.*

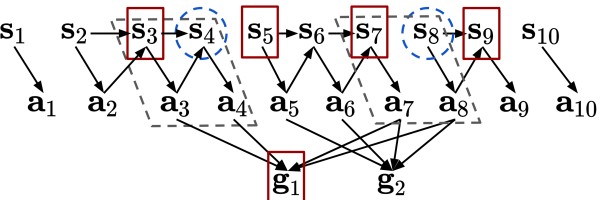

*Figure 2.* A quick example for Theorem 1. Note that the observed variables in $\mathbf{o}_t$ have been omitted for brevity. We test whether $\mathbf{S}_k = \{\mathbf{s}_3, \mathbf{s}_4\}$ and $\mathbf{S}_v = \{\mathbf{s}_7, \mathbf{s}_8\}$ belong to task $\mathbf{g}_1$ by checking the conditional dependence $\mathbf{s}_4 \not\perp\!\!\!\perp \mathbf{s}_8 \mid \mathbf{Z}_{\text{band}}(k, v, 1)$, where $\mathbf{Z}_{\text{band}}(k, v, 1) = \{\mathbf{s}_3, \mathbf{s}_5, \mathbf{s}_7, \mathbf{s}_9, \mathbf{g}_1\}$. Since $\mathbf{s}_4$ and $\mathbf{s}_8$ are conditionally dependent given $\mathbf{Z}_{\text{band}}(k, v, 1)$, $\mathbf{g}_1$ is identified as (one of) the underlying tasks. Note that our theory accommodates arbitrary disconnections between time steps (e.g., $\mathbf{s}_1$ and $\mathbf{s}_2$), multiple tasks, and arbitrarily interleaving task structures.

**Why Temporal Segmentation.** It might be worth noting that our segmentation is not about having prior knowledge of how a sequence should be divided, such as knowing in advance that a video naturally breaks into distinct semantic periods and that failing to know that will lead to a segmentation error. Instead, the purpose is simply to ensure that our tasks are well defined. The only requirement is that each segment contains more than one time step, which represents the minimal granularity needed to preserve temporal coherence. Theoretically, we could always set the segment length to minimal to capture the finests granularity of changes. In practice, one can always view the sequence as a collection of two-step segments without relying on any semantic understanding of the underlying process. With granularity this fine, segmentation has negligible effect on the tests.

Our main result is the following, with the standard Markov and Faithfulness conditions.

**Assumption 1** (Markov and Faithfulness (Spirtes et al., 2000))**.** *Let $\mathbf{G}$ be a Directed Acyclic Graph (DAG) and $\mathbb{P}$ a distribution over variables $\mathbf{V}$. The Markov property requires that each $X \in \mathbf{V}$ is independent of its nondescendants given its parents in $\mathbf{G}$. The Faithfulness requires that $\mathbb{P}$ entails no conditional independence relations beyond those implied by the Markov property of $\mathbf{G}$.*

**Theorem 1.** *Assume the Markov property and Faithfulness with respect to the graph above, and $L \geq 2$. Fix $k < v$ and a task $\mathbf{g}_i$. Then $\mathbf{g}_i$ is relevant to segments $\mathbf{S}_k$ and $\mathbf{S}_v$ if and only if*

$$\mathbf{s}_{kL} \not\perp\!\!\!\perp \mathbf{s}_{vL} \mid \mathbf{Z}_{\text{band}}(k, v, i).$$

**Proof Sketch.** The proof (Appendix A.2) relies on characterizing all possible d-connecting paths between $\mathbf{s}_{kL}$ and $\mathbf{s}_{vL}$ under the band conditioning set. Conditioning on the immediate boundary states blocks any path that propagates purely through the temporal dynamics, so dependence can only be transmitted through a shared task. Since tasks

have only incoming edges, any task other than $\mathbf{g}_i$ appears as a closed collider and blocks the path, which implies that $\mathbf{g}_i$ must be the unique source of dependence. Careful consideration of local structures and corner cases then shows that the only admissible d-connecting paths are those where actions adjacent to $\mathbf{s}_{kL}$ and $\mathbf{s}_{vL}$ both feed into $\mathbf{g}_i$.

**Implication.** Theorem 1 provides a provable way to determine whether two segments share the same task $\mathbf{g}_i$, giving an exact characterization of temporal task relevance (visualized in Fig. 2). This is powerful: once we can identify the corresponding tasks of any pair of segments, the entire task structure can be discovered. Moreover, the condition is testable directly from observed data, since conditional independence is preserved under the invertible map $\mathbf{o}_t = f_t(\mathbf{s}_t)$ and the task variables $\mathbf{g}_i$ are observed. Hence the procedure requires no parametric assumptions and is broadly applicable. Finally, the result does not rely on restrictive structural constraints, allowing tasks to appear, disappear, and interleave in arbitrary order across time, and sequences can be disconnected. This directly generalizes the most common assumption of sequential completion.

Since all states within a segment share the same task set, conditional independence (CI) tests involving the boundary states are equivalent to tests involving any other pair of states within the two segments (provided $L > 2$). Intuitively, this homogeneity means that the specific choice of representative states does not matter: any pair of states across two segments encodes the same task-level dependence. For example, $\mathbf{s}_{kL} \not\perp\!\!\!\perp \mathbf{s}_{vL} \mid \mathbf{Z}_{\mathrm{band}}(k, v, i)$ is equivalent to $\mathbf{s}_{kL-1} \not\perp\!\!\!\perp$ $\mathbf{s}_{vL-1} \mid \{\mathbf{s}_{kL-2}, \mathbf{s}_{kL}, \mathbf{s}_{vL-2}, \mathbf{s}_{vL}\} \cap \{\mathbf{s}_1, \ldots, \mathbf{s}_T\} \cup \{\mathbf{g}_i\}$. This invariance ensures that identifiability does not hinge on an arbitrary boundary choice, but is intrinsic to the task structure itself.

**Corollary 1.** *Assume the Markov property and Faithfulness with respect to the graph above, and $L > 2$. Fix $k < v$ and a task $\mathbf{g}_i$. Then $\mathbf{g}_i$ is relevant to segments $\mathbf{S}_k$ and $\mathbf{S}_v$ iff*

$$\mathbf{s}_j \not\perp\!\!\!\perp \mathbf{s}_q \mid \{\mathbf{s}_{j-1}, \mathbf{s}_{j+1}, \mathbf{s}_{q-1}, \mathbf{s}_{q+1}\} \cap \{\mathbf{s}_1, \ldots, \mathbf{s}_T\} \cup \{\mathbf{g}_i\},$$

*for any $j \in \{(k-1)L + 1, \ldots, kL\}$ and $q \in \{(v-1)L + 1, \ldots, vL\}$.*

This corollary does not impose additional conditions but establishes an equivalent characterization, guaranteed by the basic coherence of the tasks. It strengthens the applicability of Thm. 1 by showing that task relevance can be tested using arbitrary representatives within segments, not only their boundaries. Conceptually, this flexibility highlights that identifiability of the temporal task structure arises from the global dependency pattern induced by colliders, rather than from local temporal adjacency. As a consequence, the result is robust to segmentation choices and ensures that

the recovered structure reflects intrinsic properties of the underlying process rather than artifacts.

### 3.2. Discovering Global Task Structure

Building on Theorem 1 and Corollary 1, the characterization of task relevance naturally yields an algorithmic procedure. With the proposed test, one can systematically determine whether two segments share a common task. Aggregating these pairwise tests across all segment pairs yields the complete temporal task structure, as detailed in Algorithm 1.

**Proposition 1.** *Under the conditions of Theorem 1, Algorithm 1 exactly recovers the temporal task structure.*

The procedure is not only theoretically solid but also computationally efficient, which scales with the temporal horizon rather than the observation dimension. Moreover, because conditional independence is preserved under the invertible observation map, the tests can be performed directly in the observed space, without knowledge of the latent states or parametric assumptions on the dynamics. This provides an operational bridge from identifiability theory to practice: hidden temporal task structure can be precisely recovered by a simple, general, and provably correct algorithm, even in environments with arbitrary interleaving, recurrence, and disconnections across time.

---

**Algorithm 1** Global task structure discovery

---

**Input** : Segments $\mathbf{S}_{1:N}$ of length $L \geq 2$; tasks $\mathbf{G} = \{\mathbf{g}_{1:M}\}$

**Output**: Segment–task sets $\{\mathcal{T}(\mathbf{S}_k)\}_{k=1}^N$ and step labels $\{\mathcal{T}(t)\}_{t=1}^T$

$\mathcal{T}(\mathbf{S}_{1:N}) \leftarrow [\emptyset]^N; \quad \mathcal{P} \leftarrow \{(k, v) \mid 1 \leq k < v \leq N\}$

**ForEach** $i \in [1..M]$ **Do**
  **ForEach** $(k, v) \in \mathcal{P}$ **Do**
    **If** $\mathbf{s}_{kL} \not\perp\!\!\!\perp \mathbf{s}_{vL} \mid \mathbf{Z}_{\mathrm{band}}(k, v, i)$ **Then**
      $\mathcal{T}(\mathbf{S}_k) \leftarrow \mathcal{T}(\mathbf{S}_k) \cup \{\mathbf{g}_i\}; \ \mathcal{T}(\mathbf{S}_v) \leftarrow \mathcal{T}(\mathbf{S}_v) \cup \{\mathbf{g}_i\}$

**ForEach** $k \in [1..N]$ **Do**
  **ForEach** $t \in \mathbf{S}_k$ **Do** $\mathcal{T}(t) \leftarrow \mathcal{T}(\mathbf{S}_k)$

**Return** :$\{\mathcal{T}(\mathbf{S}_k)\}_{k=1}^N, \{\mathcal{T}(t)\}_{t=1}^T$

---

**What If Tasks Are Not Given.** In practice, tasks may be unobserved and must be inferred from data. In this case, we treat the inferred task representation as a latent variable and apply the CI tests to it directly, preserving the original logic. To avoid confounding, the representation is learned independently of the CI relations being evaluated. Since representation learning is conceptually separate from temporal structure recovery, extending the method to latent task settings remains fully feasible.

Moreover, the method does not require prior knowledge of the exact task set. Starting from a large pool of candidate

tasks, the algorithm provably recovers the correct subset together with its temporal structure. This assumption is substantially weaker than knowing the true task set in advance. In practice, one often has access to or can infer a broad collection of basic tasks and only needs to identify which of them, and in what structure, appear in the trajectory. Therefore, our problem setting fits a wide range of real-world scenarios even without a precise knowledge of the task set.

**Complexity.** The main practical trade-off concerns computational complexity. For large-scale datasets, using very short segment lengths leads to a large number of segments and thus many candidate temporal structures. While this does not affect correctness, the runtime grows linearly with the number of segments. In practice, increasing segment length can significantly reduce computational cost, at the expense of a modest loss in temporal resolution. This provides a controllable accuracy–scalability trade-off.

# 4. Learning Task-Relevant Representation

Having established the identifiability of temporal task structure, we now turn to the problem of learning task-relevant representations within each time step. Identifying which tasks are active at which times clarifies the dynamics across segments and ensures that temporal dependencies are properly aligned with task boundaries. This strengthens the focus on the temporal dimension, but it does not yet resolve the finer question of representation: within a single latent state $\mathbf{s}_t$, only a subset of variables may be relevant to the task, while the rest correspond to nuisance factors. To obtain a minimal yet sufficient representation, we must therefore dig deeper into the latent space of $\mathbf{s}_t$ and disentangle the components that are truly task-relevant from those that are irrelevant. Specifically, we aim to ensure that the estimated latents (e.g., $\mathbf{s}_{t_1}$) associated with each task (e.g., $t_1$) are not functions of any other latent variables, whether tied to other tasks or unrelated altogether

Identifiability of the latent variables concerns recovering the unique ground truth $\mathbf{s}_t$ from two observationally equivalent models $\mathbf{o}_t = f_t(\mathbf{s}_t)$ and $\mathbf{o}_t = \hat{f}_t(\hat{\mathbf{s}}_t)$. Let $\mathbf{g} = u(\mathbf{s}, \theta)$ and $\hat{\mathbf{g}} = \hat{u}(\hat{\mathbf{s}}, \hat{\theta})$, where $\theta$ and $\hat{\theta}$ denote variables other than $\mathbf{s}$ and $\hat{\mathbf{s}}$. These mappings exist due to the dependency structure $\mathbf{s}_t \rightarrow \mathbf{a}_t \rightarrow \mathbf{g}_i$. With slight abuse of notation, we mostly omit $\theta$ and $\hat{\theta}$ and write $\mathbf{g} = u(\mathbf{s})$ and $\hat{\mathbf{g}} = \hat{u}(\hat{\mathbf{s}})$ for brevity.

## 4.1. Identifiability with a Generalist Model

We begin by asking what can be achieved without imposing any structural constraint beyond observational equivalence. That is, we consider a *generalist model* without explicitly being regularized to focus on the corresponding tasks. While such a model may capture the necessary information for prediction, its ability to recover the ground-truth task–relevant latent representation is limited.

**Additional Notation.** For a vector-valued function $u : \mathbb{R}^{d_s} \rightarrow \mathbb{R}^{d_g}$, we denote by $J_u(\mathbf{s}_t)$ the Jacobian matrix with respect to $\mathbf{s}_t$, whose $(i, j)$ entry is $\partial u_i / \partial s_{t,j}$. For a vector or matrix $A$, we write $\mathcal{I}(A)$ for the set of indices corresponding to its nonzero entries, and $\|\mathcal{I}(A)\|$ for its cardinality (the number of nonzeros, i.e., the $\ell_0$ norm). We denote $I_k \subseteq [d_s]$ as the set of indices of the latent variables relevant to task $g_k$, and $\mathbf{s}_{t,I_k}$ as the corresponding set of latent variables.

**Proposition 2.** *Assume that, for each $i \in [d_g]$, there exists a set $\mathcal{N}_i$ of $\|\mathcal{I}(J_u(\mathbf{s}_t)_{i,\cdot})\|$ distinct points such that the corresponding Jacobian row vectors*

$$\left( \frac{\partial u_i}{\partial s_{t,1}}, \frac{\partial u_i}{\partial s_{t,2}}, \ldots, \frac{\partial u_i}{\partial s_{t,d_s}} \right) \Big|_{\mathbf{s}_t = \mathbf{s}_t^{(l)}}, \quad l \in \mathcal{N}_i,$$

*are linearly independent, and $\mathcal{I}\left( (J_u(\mathbf{s}_t^{(l)}) M)_{i,\cdot} \right) \subseteq \mathcal{I}\left( (J_{\hat{u}}(\hat{\mathbf{s}}_t^{(l)}))_{i,\cdot} \right)$, where $M$ is a matrix sharing the nonzero index set of matrix-valued function $M'(\mathbf{s}, \hat{\mathbf{s}})$ in $J_u(\mathbf{s}) M'(\mathbf{s}, \hat{\mathbf{s}}) = J_{\hat{u}}(\hat{\mathbf{s}})$. Then, for any task $\mathbf{g}_k$ with latent index set $I_k$, the number of estimated task-relevant latent variables is larger than that of the ground truth, i.e.,*

$$\|\mathcal{I}\left( (J_{\hat{u}})_{i,\cdot} \right)\| \geq \|\mathcal{I}\left( (J_u)_{i,\cdot} \right)\|.$$

**Proof Sketch.** The argument starts by connecting the support of the Jacobian to the underlying dependency graph. The span condition ensures that the information is being preserved during estimation, and thus no true dependency can be eliminated in the transformation between $\mathbf{s}$ and $\hat{\mathbf{s}}$. Equivalently, the nonzero pattern of $J_u(\mathbf{s})$ must be contained within that of $J_{\hat{u}}(\hat{\mathbf{s}})$. Translated back to the task–latent structure, this implies that the number of the estimated task-relevant latent variables, as captured by the support, is always a superset of the true one.

**Discussion on Assumptions.** The requirement of sufficient nonlinearity is standard in identifiability analyses of nonlinear models (Lachapelle et al., 2022; Zheng et al., 2022). Specifically, it rules out degenerate cases where samples concentrate on an extremely small subset (e.g., as few as several samples) such that the Jacobian vectors cannot even span their own supports. At the same time, identifiability is defined as an asymptotic property (infinite samples), and the assumption only requires the existence of several nondegenerate samples in the whole space, which is almost always satisfied in practice. More detailed discussion on the assumption is in Appendix B.

**Implication.** This result shows that, without explicit modelling of specific tasks, generalist models tend to learn a representation that is larger than necessary. The conclusion is intuitive: a sufficiently expressive foundation model can capture a representation that contains all information needed for downstream tasks.

At the same time, the guarantee $\|\mathcal{I}(J_{\hat{u}})_{i,\cdot}\| \geq \|\mathcal{I}(J_u)_{i,\cdot}\|$ remains weak: it ensures only that the estimated representation is no smaller in *size* than the true one, not that it matches it or recovers the correct variables. In practice, this means a generalist model often learns an overcomplete representation, where task-relevant variables may still be missed or entangled with irrelevant ones. Worse, the inequality provides no guarantee on recovering variable values since the enlarged representation may distort or discard essential information. This formalizes the intuition that while frontier generalist models are expressive enough to encode all task information, without additional regularization they fail to isolate the minimal task-relevant latent representation.

### 4.2. From Generalist to Specialist

The previous result shows that a generalist model often produces an enlarged representation, estimating more task-relevant variables than truly exist. Such oversizing does not guarantee that all genuine factors are preserved; irrelevant latents may be included, while essential ones can still be distorted or obscured. To recover the *true* task-relevant representation, additional inductive bias is needed.

A natural choice is sparsity regularization on the estimated task–latent structure, which enforces minimality in the recovered structure. Intuitively, sparsity prunes away superfluous dimensions and curbs over-expansion, ensuring that the final representation retains only the variables genuinely required for each task. The corresponding gurantee is as follows:

**Theorem 2.** *Consider two observationally equivalent generative processes* $\mathbf{o}_t = f_t(\mathbf{s}_t)$ *and* $\mathbf{o}_t = \hat{f}_t(\hat{\mathbf{s}}_t)$, *and assume the conditions in Proposition 2. Then, for any task* $\mathbf{g}_k$ *with latent index set* $I_k$, *with a sparsity regularization*

$$\|\mathcal{I}(J_{\hat{u}})\| \leq \|\mathcal{I}(J_u)\|,$$

*under some permutation* $\pi$, *the estimated task-relevant latent variables* $\hat{\mathbf{s}}_{t,\pi(I_k)}$ *are an invertible function* $h_k$ *of only the ground-truth task-relevant latent variables* $\mathbf{s}_{t,I_K}$, *i.e.,*

$$\hat{\mathbf{s}}_{t,\pi(I_k)} = h_k(\mathbf{s}_{t,I_K}).$$

**Proof Sketch.** Under the span assumptions from Proposition 2, we first show that every nonzero entry of $J_u(\mathbf{s})$ must correspond to a nonzero entry of $J_{\hat{u}}(\hat{\mathbf{s}})$, up to a column permutation $\pi$. Sparsity regularization enforces that no additional entries can remain nonzero, which upgrades inclusion into exact equivalence of supports. Finally, algebraic analysis helps move from structure to variables, exploiting the separation between task-relevant and task-irrelevant latents.

**Implication.** *Practically*, Theorem 2 establishes the formal guarantees on going from a generalist to a specialist model. It shows that, based on the general guarantee in Proposition 2, a simple sparsity regularization is sufficient to disentangle task-relevant latent variables from the irrelevant ones. Unlike the generalist guarantee, which recovers a superset of the true support, the sparsity constraint sharpens recovery to the variable-level and disentangles irrelevant parts. Conceptually, this result highlights the necessity of moving from generalist to specialist modeling: while a generalist can cover all possible dependencies, only task-specific modeling with appropriate regularization yields a representation that is both minimal and faithful. This provides a principled justification for why specialist models can achieve disentangled task representations where generalist models cannot, offering formal guarantees for the intuition.

*Theoretically*, our result suggests a new strategy for provably uncovering the latent variables underlying the observational world. Importantly, because we allow arbitrary disconnections between time steps, Theorem 2 also covers the i.i.d. setting. This is a substantially harder case than prior work that exploits temporal information or domain shifts, since changes across time or environments inherently provide extra signals for identification, whereas identifiability in the absence of changes is notoriously difficult. Existing i.i.d. results rely either on restrictive functional assumptions (Taleb & Jutten, 1999; Buchholz et al., 2022) or graphical criteria on the underlying structure (Moran et al., 2021; Zheng et al., 2022) to achieve full component-wise identifiability. By contrast, our focus is not on recovering every individual latent, but rather on identifying all task-relevant ones as a subgroup. This relaxation allows us to bypass such strong assumptions and still establish general identifiability guarantees. Beyond our setting, this perspective may prove methodologically useful for a wide range of latent-variable problems, where isolating task-relevant latents is important.

## 5. Experiments

In this section, we present comprehensive empirical results supporting our theory on both temporal task structure learning and task-relevant representation learning across diverse settings. Due to page limits, some setup details are deferred to Appendix C, and we have included many additional empirical results in Appendix D.

**Identifiability of Temporal Task Structure.** We evaluate whether the proposed algorithm can recover temporal task structures under challenging conditions, including disconnected temporal relations and arbitrarily interleaving tasks. Two setups are considered: (1) varying the number of time steps $T$ from 8 to 20 with $T/5$ tasks, and (2) varying the number of tasks $M$ from 2 to 10 with 20 time steps. To maximize structural complexity, we set the minimum segment length to 2 and randomly generate the task–time step dependencies. Additionally, 20% of the dependencies between

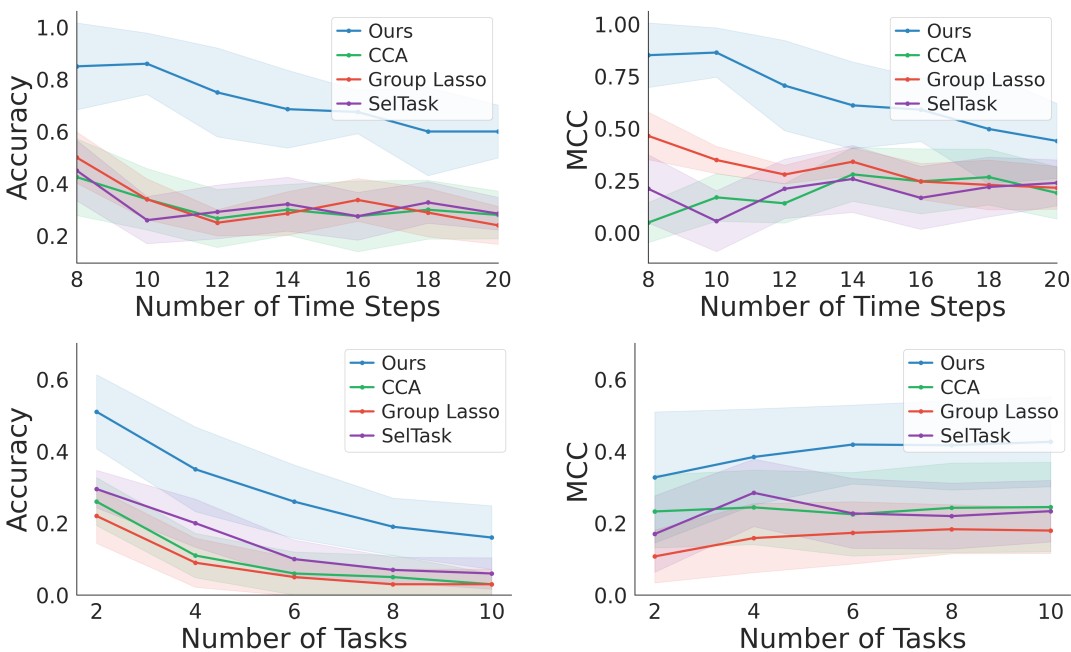

*Figure 3.* Temporal task structure identification. Top: varying the number of time steps $T$ with $T/5$ tasks. Bottom: varying the number of tasks $M$ with 20 time steps. Left: accuracy. Right: MCC.

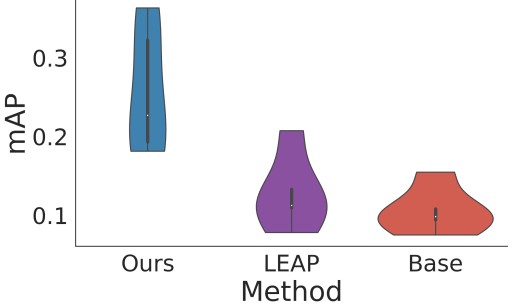

*Figure 4.* Real-world temporal task structure discovery.

consecutive segments are randomly removed. Each dataset contains $10,000$ samples generated from linear Gaussian SCMs. All results are from 10 random runs with Fisher's z-test (Fisher, 1921) with the p-value threshold as 0.05.

For both settings, we report accuracy and Matthews correlation coefficient (MCC) of the tasks identified. For baselines, we consider classical CCA (Anderson, 2003) and Group Lasso (Yuan & Lin, 2006), as well as the most recent SelTask (Qiu et al., 2024). Results are summarized in Fig. 3. Ours dominates across all $T$ and $M$ in both accuracy and MCC. As expected, performance degrades as the problem becomes harder (larger $T$ or $M$), but the gap persists.

**Real-World Structure.** To explore whether we can identify real-world structures between tasks and time steps, we further conduct experiments on the recent SportsHHI video dataset (Wu et al., 2024). For each time frame in the video, the objective is to discover its corresponding task labels,

which in this context correspond to the behaviors of humans captured in the video. Because the dataset involves multiple individuals with complex and overlapping interactions, each frame typically contains multiple task labels, resulting in highly intricate task structures. This makes it a challenging and suitable testbed for stress-testing the identification.

Following common practice, we use a pretrained CLIP encoder (Radford et al., 2021) to obtain visual embeddings $\mathbf{o}$ and task embeddings $\mathbf{g}$, and employ a variational autoencoder to estimate the latent state variables $\mathbf{s}$. The latent transition dynamics between consecutive states are parameterized by an MLP, while conditional mutual information (CMI) is used as a proxy for conditional independence to mitigate the curse of dimensionality in statistical testing. We compare our approach against two baselines: (i) applying Alg. 1 directly to observed variables instead of latent ones (replacing $\mathbf{s}$ with $\mathbf{o}$), and (ii) LEAP (Yao et al., 2021), a representative method for learning latent temporal representations with identifiability guarantees on the latent variables but *not* on the structure. Following prior work, we evaluate using mean average precision (mAP). The results in Fig. 4 demonstrate that modeling the complexity of general temporal task structures is essential for accurate discovery in complex real-world scenarios. It might be worth noting that we have also compared with more standard video models that do not target identiability, of which the results are shown in Table 3 in Appendix D.

**Identifiability of Task-Relevant Representation.** After establishing identifiability of the temporal task structure,

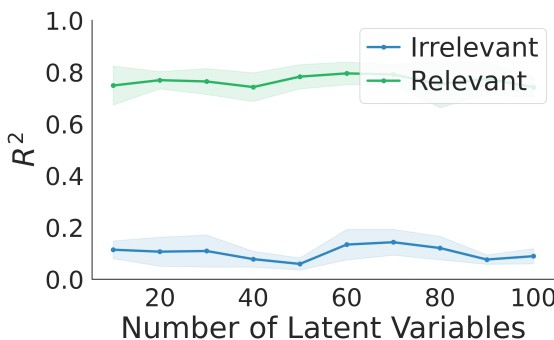

*Figure 5.* $R^2$ for relevant and irrelevant parts.

we zoom in on a single step and evaluate recovery of the task-relevant latent representation conditioned on the corresponding tasks. The data-generating process follows the theoretical setup, with an MLP using Leaky ReLU as the nonlinear function. For every dataset, we randomly select $1/5$ dimensions as task-relevant latent variables. For estimation, we employ a VAE with $\ell_1$ regularization on the task-latent structure. As the evaluation metric, we report the $R^2$ between the estimated and ground-truth latent components: higher values indicate accurate recovery of relevant parts, while lower values indicate effective separation from irrelevant parts. Figure 5 shows a clear gap: (1) task-relevant representations are successfully disentangled from irrelevant ones (low $R^2$ for irrelevant parts), and (2) the estimated task-relevant part captures most of the information in the ground-truth one (high $R^2$ for relevant parts). These provide rigorous validation of the identifiability theory, confirming that task-relevant latent variables can be uncovered as a group with both information preservation and irrelevance disentanglement in practice.

**Task-Relevant Latents in Realistic Vision.** We next investigate the recovery of task-relevant latent representations in realistic scenarios. Since ground-truth latents are usually unavailable in practice, direct comparison with the truth is infeasible. Evaluation therefore turns to human interpretability, where visualizing the identified latents provides key evidence. We construct a dataset of cat images using Flux, with tasks such as "wearing eyeglasses," "wearing a hat," and "wearing a tie," explicitly considering realistic images to align with real-world vision. For estimation, we adopt a GAN-based generator where each task is associated with a learned transformation of the latents. Concretely, given $\mathbf{s}$, a task-specific operator modifies only a sparse subset of coordinates by an $\ell_1$-regularized mask, producing masked latents that are then passed to the generator. Figure 6 compares results with and without sparsity. With sparsity, the recovered latents correspond closely to the intended task attributes, while without sparsity, irrelevant factors such as color are entangled with the target tasks. This further supports the task-relevant identifiability and the role of sparsity. Additional results on more scenarios are included in

With sparsity

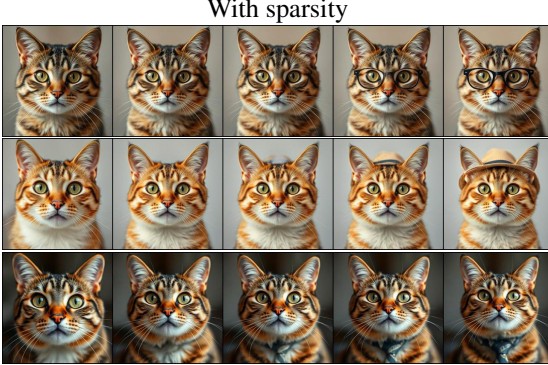

Without sparsity

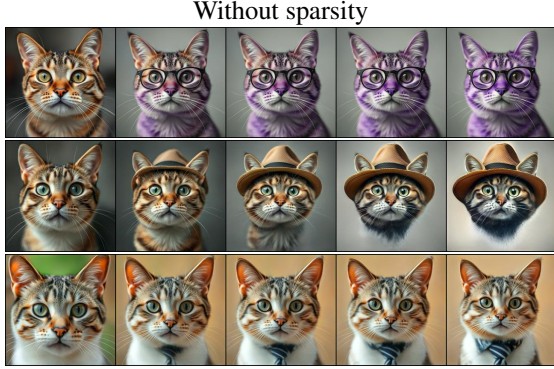

*Figure 6.* Qualitative comparison of identified task-relevant latents. Tasks include "wearing glasses," "wearing a hat," and "wearing a tie." With sparsity, the model isolates a minimal but sufficient subset of latents aligned with each task. Without sparsity, irrelevant factors such as color become entangled with task-relevant ones.

Appendix D (e.g., Figure 7), which further verify the claims.

## 6. Conclusion and Discussion

In this paper, we initiated the theoretical investigation of learning task-relevant world representations, aiming to move from generalist to specialist. The main challenges lie in the level of generality, which requires handling both complex structures, such as disconnected sequences, interleaving tasks, and frequent switches, and general processes, including nonlinear functions, arbitrary distributions, and the absence of auxiliary information. While we have addressed these, several related questions remain open. First, although identifiability is defined asymptotically and frontier models are often trained on web-scale data, it is still important to understand the finite-sample regime, and the lack of related analysis is a limitation in data-sparse scenarios. Second, our present way of leveraging identifiability is relatively simple, essentially a standard estimator with sparsity regularization. While such simplicity and universality are often advantageous, it is also intriguing to consider identifiability-inspired architectures that depart more radically from existing patterns. A stronger focus on identifiability within the community may reveal barrier-breaking insights that have been overshadowed by the pursuit of purely empirical gains, and we aim to contribute toward this shift.

## Impact Statement

This paper presents work whose goal is to advance the field of Machine Learning. There are many potential societal consequences of our work, none which we feel must be specifically highlighted here.

## Acknowledgment

The authors would like to thank the anonymous reviewers for helpful comments and suggestions. The authors would also like to acknowledge the support from NSF Award No. 2229881, AI Institute for Societal Decision Making (AI-SDM), the National Institutes of Health (NIH) under Contract R01HL159805, and grants from Quris AI, Florin Court Capital, MBZUAI-WIS Joint Program, and the Al Deira Causal Education project.

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

# Appendices

## Table of Contents

# A. Proofs

## A.1. Notation

We first provide a summary of notation in Table 1.

| Symbol | Meaning |
|---|---|
| $o_t \in \mathbb{R}^{d_o}$ | Observation at time $t$ |
| $s_t \in \mathbb{R}^{d_s}$ | Latent state at time $t$ |
| $a_t \in \mathbb{R}^{d_a}$ | Action at time $t$ |
| $g_i$ | Task variable $i$, defined as collider across time steps |
| $M$ | Total number of tasks |
| $T$ | Total number of time steps |
| $S_k$ | Segment $k$, a block of consecutive latent steps |
| $T(t)$ | Set of tasks relevant to time step $t$ |
| $T(S_k)$ | Set of tasks relevant to segment $S_k$ |
| $f_t$ | Observation function $s_t \mapsto o_t$, diffeomorphism onto image |
| $\pi_t$ | Action policy $s_t \mapsto a_t$ with noise $\eta_t$ |
| $F_t$ | Transition function for connected boundaries |
| $F_t'$ | Transition function for disconnected boundaries |
| $J_u(s_t)$ | Jacobian of mapping $u$ w.r.t. latent state $s_t$ |
| $I(A)$ | Index set of nonzero entries of matrix/vector $A$ |
| $\|I(A)\|$ | Cardinality of index set $I(A)$ (i.e., $\ell_0$ norm) |
| $I_k \subseteq [d_s]$ | Index set of latents relevant to task $g_k$ |
| $s_{t,I_k}$ | Latent variables in $s_t$ relevant to $g_k$ |

*Table 1.* Summary of notation.

## A.2. Proof of Theorem 1

**Theorem 1.** *Assume the Markov property and Faithfulness with respect to the graph above, and $L \geq 2$. Fix $k < v$ and a task $\mathbf{g}_i$. Then $\mathbf{g}_i$ is relevant to segments $\mathbf{S}_k$ and $\mathbf{S}_v$ if and only if*

$$\mathbf{s}_{kL} \not\perp\!\!\!\perp \mathbf{s}_{vL} \,\Big|\, \mathbf{Z}_{\text{band}}(k, v, i).$$

**Notation and Blocking Rules.**  A path is a sequence of distinct nodes $(v_0, \ldots, v_r)$ with each consecutive pair adjacent. Along a path, a node is a collider if both incident path edges have arrowheads into the node, and a non-collider otherwise. A path is blocked by a conditioning set $\mathbf{Z}$ if it contains a non-collider in $\mathbf{Z}$ or a collider that is neither in $\mathbf{Z}$ nor has a descendant in $\mathbf{Z}$ (i.e., d-separation (Pearl, 1988)). In our graph, tasks have only incoming edges, and tasks have no descendants.

**Band Conditioning Set.**  Throughout this subsection the conditioning set is

$$\mathbf{Z}_{\text{band}}(k, v, i) = \{\mathbf{s}_{kL-1}, \mathbf{s}_{kL+1}, \mathbf{s}_{vL-1}, \mathbf{s}_{vL+1}\} \cap \{\mathbf{s}_1, \ldots, \mathbf{s}_T\} \cup \{\mathbf{g}_i\}, \tag{4}$$

with out-of-range indices omitted. Thus only the two immediate inner neighbors $\mathbf{s}_{kL+1}, \mathbf{s}_{vL-1}$ and the two immediate outer neighbors $\mathbf{s}_{kL-1}, \mathbf{s}_{vL+1}$ (when they exist) are conditioned; among tasks only $\mathbf{g}_i$ is conditioned.

**Lemma 1** (A d-connecting path uses exactly one task, equal to $\mathbf{g}_i$). *Every path from $\mathbf{s}_{kL}$ to $\mathbf{s}_{vL}$ that is d-connecting given $\mathbf{Z}_{\text{band}}(k, v, i)$ contains exactly one task node, and that task is $\mathbf{g}_i$.*

*Proof.* Consider any path with no task nodes. Such a path alternates among states and actions and moves in time via $s_t \to s_{t+1}$, $s_t \to a_t$ or $a_t \to s_{t+1}$. Any forward traversal from $\mathbf{s}_{kL}$ toward $\mathbf{s}_{vL}$ must pass through the cut state $\mathbf{s}_{kL+1}$; symmetrically, any approach into $\mathbf{s}_{vL}$ from the left must pass through $\mathbf{s}_{vL-1}$. All these cut states are in $\mathbf{Z}_{\text{band}}$ and are non-colliders on such chain paths, so the path is blocked. Hence, any d-connected path must include at least one task.

If a path contains a task $g_j \neq \mathbf{g}_i$, then at $g_j$ both incident edges point into $g_j$, so $g_j$ is a collider. Since $g_j \notin \mathbf{Z}_{\text{band}}$ and tasks have no descendants, this collider is closed and the path is blocked. Therefore no d-connecting path can contain any task other than $\mathbf{g}_i$.

If a path contains two or more tasks, at least one of them is not $\mathbf{g}_i$, which blocks the path by the previous argument. Thus every d-connecting path contains exactly one task and that task is $\mathbf{g}_i$. $\qquad\square$

**Lemma 2** (Local structure of d-connecting paths). *Under the graph and conditioning in Equation 4, every d-connecting path between $\mathbf{s}_{kL}$ and $\mathbf{s}_{vL}$ has one of the four forms*

$$
\begin{array}{ll}
\textit{(I)} & \mathbf{s}_{kL} \to \mathbf{a}_{kL} \to \mathbf{g}_i \leftarrow \mathbf{a}_{vL} \leftarrow \mathbf{s}_{vL}, \qquad \textit{(II)} \quad \mathbf{s}_{kL} \to \mathbf{a}_{kL} \to \mathbf{g}_i \leftarrow \mathbf{a}_{vL-1} \to \mathbf{s}_{vL}, \\
\textit{(III)} & \mathbf{s}_{kL} \leftarrow \mathbf{a}_{kL-1} \to \mathbf{g}_i \leftarrow \mathbf{a}_{vL} \leftarrow \mathbf{s}_{vL}, \qquad \textit{(IV)} \quad \mathbf{s}_{kL} \leftarrow \mathbf{a}_{kL-1} \to \mathbf{g}_i \leftarrow \mathbf{a}_{vL-1} \to \mathbf{s}_{vL},
\end{array}
\tag{5}
$$

*with out-of-range indices omitted.*

*Proof.* By Lemma 1, any d-connecting path contains exactly the single task $\mathbf{g}_i$.

*Left boundary.* The first neighbor of $\mathbf{s}_{kL}$ on any d-connecting path cannot be a state, because the only state neighbors are $\mathbf{s}_{kL-1}$ and $\mathbf{s}_{kL+1}$, both in $\mathbf{Z}_{\text{band}}$ and both non-colliders on chain moves, which would block the path. Hence the neighbor must be an adjacent action, $\mathbf{a}_{kL-1}$ (if $kL > 1$) or $\mathbf{a}_{kL}$. From that action, any continuation to a state would encounter one of the conditioned cut states as a non-collider, so the next node must be $\mathbf{g}_i$ via an edge $a \to \mathbf{g}_i$. This yields the two left fragments $\mathbf{s}_{kL} \leftarrow \mathbf{a}_{kL-1} \to \mathbf{g}_i$ and $\mathbf{s}_{kL} \to \mathbf{a}_{kL} \to \mathbf{g}_i$.

*Right boundary.* Symmetrically, the predecessor of $\mathbf{s}_{vL}$ on the path cannot be a state, since the only state neighbors are $\mathbf{s}_{vL-1}$ and $\mathbf{s}_{vL+1}$, which are in $\mathbf{Z}_{\text{band}}$ and would block as non-colliders in the potential additional paths. Specifically, $\mathbf{s}_{vL-1}$ is a non-collider in paths involving $\mathbf{s}_{vL-1} \to \mathbf{s}_{vL}$, and $\mathbf{s}_{vL+1}$ is a non-collider in paths involving $\mathbf{s}_{vL+1} \to \mathbf{a}_{vL+1}$ or $\mathbf{s}_{vL+1} \to \mathbf{s}_{vL+2}$. Thus the predecessor must be $\mathbf{a}_{vL-1}$ or $\mathbf{a}_{vL}$, linked to $\mathbf{g}_i$ by an edge $\mathbf{a} \to \mathbf{g}_i$ traversed in reverse on the path. This yields the two right fragments $\mathbf{g}_i \leftarrow \mathbf{a}_{vL-1} \to \mathbf{s}_{vL}$ and $\mathbf{g}_i \leftarrow \mathbf{a}_{vL} \leftarrow \mathbf{s}_{vL}$.

Combining the two left with the two right fragments gives exactly the four forms in Equation 5. On each such path, $\mathbf{g}_i$ is the unique collider and is in $\mathbf{Z}_{\text{band}}$, while all other nodes are non-colliders that are not in $\mathbf{Z}_{\text{band}}$, so these paths are d-connecting. $\qquad\square$

Now we are ready to prove the theorem.

**Theorem 1.** *Assume the Markov property and Faithfulness with respect to the graph above, and $L \geq 2$. Fix $k < v$ and a task $\mathbf{g}_i$. Then $\mathbf{g}_i$ is relevant to segments $\mathbf{S}_k$ and $\mathbf{S}_v$ if and only if*

$$
\mathbf{s}_{kL} \not\perp\!\!\!\perp \mathbf{s}_{vL} \,\Big|\, \mathbf{Z}_{\text{band}}(k, v, i).
$$

*Proof.* ($\Rightarrow$) Suppose $\mathbf{s}_{kL}$ and $\mathbf{s}_{vL}$ are conditionally dependent given $\mathbf{Z}_{\text{band}}(k, v, i)$. By Lemma 2, there exists a d-connecting path of one of the four forms in Equation 5. In each form, the actions adjacent to $\mathbf{s}_{kL}$ and $\mathbf{s}_{vL}$ that appear on the path are parents of $\mathbf{g}_i$. Hence $\mathbf{g}_i$ is relevant to segments $\mathbf{S}_k$ and $\mathbf{S}_v$.

($\Leftarrow$) Conversely, suppose both intersections are nonempty. Choose $p \in \{kL - 1, kL\}$ and $q \in \{vL - 1, vL\}$ such that $\mathbf{a}_p \to \mathbf{g}_i$ and $\mathbf{a}_q \to \mathbf{g}_i$. Then one of the four forms in Equation 5 exists. Along that path, $\mathbf{g}_i$ is the unique collider and is conditioned, while all other nodes are non-colliders not in $\mathbf{Z}_{\text{band}}(k, v, i)$. Therefore the path is not blocked and

$$
\mathbf{s}_{kL} \not\perp\!\!\!\perp \mathbf{s}_{vL} \,\Big|\, \mathbf{Z}_{\text{band}}(k, v, i).
\tag{6}
$$

This proves the equivalence stated in Theorem 1. $\qquad\square$

### A.3. Proof of Corollary 1

**Corollary 1.** *Assume the Markov property and Faithfulness with respect to the graph above, and $L > 2$. Fix $k < v$ and a task $\mathbf{g}_i$. Then $\mathbf{g}_i$ is relevant to segments $\mathbf{S}_k$ and $\mathbf{S}_v$ iff*

$$
\mathbf{s}_j \not\perp\!\!\!\perp \mathbf{s}_q \,\Big|\, \{\mathbf{s}_{j-1}, \mathbf{s}_{j+1}, \mathbf{s}_{q-1}, \mathbf{s}_{q+1}\} \cap \{\mathbf{s}_1, \ldots, \mathbf{s}_T\} \cup \{\mathbf{g}_i\},
$$

*for any $j \in \{(k-1)L + 1, \ldots, kL\}$ and $q \in \{(v-1)L + 1, \ldots, vL\}$.*

*Proof.* Fix $k < v$, pick any $j \in \{(k-1)L+1, \ldots, kL\}$ and $q \in \{(v-1)L+1, \ldots, vL\}$, and define the local band set

$$\mathbf{Z}_{\mathrm{loc}}(j,q,i) = \{\mathbf{s}_{j-1}, \mathbf{s}_{j+1}, \mathbf{s}_{q-1}, \mathbf{s}_{q+1}\} \cap \{\mathbf{s}_1, \ldots, \mathbf{s}_T\} \cup \mathbf{g}_i. \tag{7}$$

By the same blocking argument as in Lemma 1, any d-connecting path between $\mathbf{s}_j$ and $\mathbf{s}_q$ given $\mathbf{Z}_{\mathrm{loc}}(j,q,i)$ must contain exactly one task node and it must be $\mathbf{g}_i$. The neighbor of $\mathbf{s}_j$ on any such path cannot be a state, since $\mathbf{s}_{j-1}$ and $\mathbf{s}_{j+1}$ are in $\mathbf{Z}_{\mathrm{loc}}$ and are non-colliders on chain moves, hence they would block. Therefore the path must leave $\mathbf{s}_j$ through an adjacent action $\mathbf{a}_{j-1}$ or $\mathbf{a}_j$, and from there enter $\mathbf{g}_i$ via an edge $\mathbf{a} \to \mathbf{g}_i$. A symmetric argument holds at the right end near $\mathbf{s}_q$. Consequently every d-connecting path between $\mathbf{s}_j$ and $\mathbf{s}_q$ given $\mathbf{Z}_{\mathrm{loc}}(j,q,i)$ has one of the four forms

$$
\begin{aligned}
&\text{(I) } \mathbf{s}_j \to \mathbf{a}_j \to \mathbf{g}_i \leftarrow \mathbf{a}_q \leftarrow \mathbf{s}_q, \qquad &&\text{(II) } \mathbf{s}_j \to \mathbf{a}_j \to \mathbf{g}_i \leftarrow \mathbf{a}_{q-1} \to \mathbf{s}_q, \\
&\text{(III) } \mathbf{s}_j \leftarrow \mathbf{a}_{j-1} \to \mathbf{g}_i \leftarrow \mathbf{a}_q \leftarrow \mathbf{s}_q, \qquad &&\text{(IV) } \mathbf{s}_j \leftarrow \mathbf{a}_{j-1} \to \mathbf{g}_i \leftarrow \mathbf{a}_{q-1} \to \mathbf{s}_q,
\end{aligned}
\tag{8}
$$

with out-of-range indices omitted. On each such path $\mathbf{g}_i$ is the unique collider and is conditioned, while all other nodes are non-colliders that are not conditioned, so the path is d-connecting.

Note that states inside the same segment share the same task set, and task nodes have only incoming edges from actions. It follows that, if $\mathbf{g}_i$ is relevant to $\mathbf{S}_k$ and $\mathbf{S}_v$ then there exist $p \in j-1, j$ and $r \in q-1, q$ such that $\mathbf{a}_p \to \mathbf{g}_i$ and $\mathbf{a}_r \to \mathbf{g}_i$.

Then we prove the equivalence as follows.

($\Rightarrow$) If $\mathbf{g}_i$ is relevant to $\mathbf{S}_k$ and $\mathbf{S}_v$, pick $p \in j-1, j$ and $r \in q-1, q$ with $\mathbf{a}_p \to \mathbf{g}_i$ and $\mathbf{a}_r \to \mathbf{g}_i$ as above. Then one of the four local forms in Equation 8 exists and is d-connecting given $\mathbf{Z}_{\mathrm{loc}}(j,q,i)$, hence

$$\mathbf{s}_j \not\perp\!\!\!\perp \mathbf{s}_q \mid \mathbf{Z}_{\mathrm{loc}}(j,q,i). \tag{9}$$

($\Leftarrow$) Conversely, if $\mathbf{s}_j$ and $\mathbf{s}_q$ are conditionally dependent given $\mathbf{Z}_{\mathrm{loc}}(j,q,i)$, then by Equation 8 the actions adjacent to $\mathbf{s}_j$ and $\mathbf{s}_q$ that lie on a d-connecting path are parents of $\mathbf{g}_i$. Together with the segment homogeneity, $\mathbf{g}_i$ is relevant to $\mathbf{S}_k$ and $\mathbf{S}_v$.

This proves the stated equivalence for arbitrary $j \in \mathbf{S}_k$ and $q \in \mathbf{S}_v$ with $L > 2$. $\qquad \square$

## A.4. Proof of Proposition 1

**Proposition 1.** *Under the conditions of Theorem 1, Algorithm 1 exactly recovers the temporal task structure.*

*Proof.* Fix a task $\mathbf{g}_i$ and a segment $\mathbf{S}_k$. If $\mathbf{S}_k$ truly contains $\mathbf{g}_i$, then for $\mathbf{S}_v$ with $v \neq k$ that also contains $\mathbf{g}_i$, by Theorem 1, there must be

$$\mathbf{s}_{kL} \not\perp\!\!\!\perp \mathbf{s}_{vL} \mid \mathbf{Z}_{\mathrm{band}}(k,v,i). \tag{10}$$

The oracle CI test returns dependence, so Algorithm 1 adds $\mathbf{g}_i$ to both $\mathcal{T}(\mathbf{S}_k)$ and $\mathcal{T}(\mathbf{S}_v)$.

Conversely, if $\mathcal{T}(\mathbf{S}_k)$ does not contain $\mathbf{g}_i$, then Theorem 1 implies conditional independence for all pairs involving $\mathbf{S}_k$, hence the algorithm never adds $\mathbf{g}_i$ to $\mathcal{T}(\mathbf{S}_k)$. Therefore the recovered segment–task incidence is exact. Per-step labels are correct by assignment. $\qquad \square$

## A.5. Proof of Proposition 2

**Proposition 2.** *Assume that, for each $i \in [d_g]$, there exists a set $\mathcal{N}_i$ of $\|\mathcal{I}(J_u(\mathbf{s}_t)_{i,\cdot})\|$ distinct points such that the corresponding Jacobian row vectors*

$$\left( \frac{\partial u_i}{\partial s_{t,1}}, \frac{\partial u_i}{\partial s_{t,2}}, \ldots, \frac{\partial u_i}{\partial s_{t,d_s}} \right) \Bigg|_{\mathbf{s}_t = \mathbf{s}_t^{(l)}}, \quad l \in \mathcal{N}_i,$$

*are linearly independent, and $\mathcal{I}\left( (J_u(\mathbf{s}_t^{(l)}) M)_{i,\cdot} \right) \subseteq \mathcal{I}\left( (J_{\hat{u}}(\hat{\mathbf{s}}_t^{(l)}))_{i,\cdot} \right)$, where $M$ is a matrix sharing the nonzero index set of matrix-valued function $M'(\mathbf{s}, \hat{\mathbf{s}})$ in $J_u(\mathbf{s}) M'(\mathbf{s}, \hat{\mathbf{s}}) = J_{\hat{u}}(\hat{\mathbf{s}})$. Then, for any task $\mathbf{g}_k$ with latent index set $I_k$, the number of estimated task-relevant latent variables is larger than that of the ground truth, i.e.,*

$$\|\mathcal{I}\left( (J_{\hat{u}})_{i,\cdot} \right)\| \geq \|\mathcal{I}\left( (J_u)_{i,\cdot} \right)\|.$$

*Proof.* Since $\mathbf{o}_t = f_t(\mathbf{s}_t)$ and $o = \hat{f}_t(\hat{\mathbf{s}}_t)$ are observationally equivalent, there exists an invertible mapping $\phi$ such that

$$\hat{\mathbf{s}}_t = \hat{f}_t^{-1} \circ f_t(\mathbf{s}_t) = \phi(\mathbf{s}_t), \tag{11}$$

with inverse $\phi^{-1}$. By the chain rule,

$$J_{\hat{u}} = J_u J_{\phi^{-1}}. \tag{12}$$

Fix $i \in [d_g]$. Consider the set $\mathcal{N}_i$ of $\|\mathcal{I}((J_u)_{i,\cdot})\|$ distinct points and the corresponding Jacobian row vectors

$$\left( \frac{\partial u_i}{\partial \mathbf{s}_{t,1}}, \ldots, \frac{\partial u_i}{\partial s_{t,d_s}} \right) \Big|_{\mathbf{s}_t = \mathbf{s}_t^{(l)}}, \quad l \in \mathcal{N}_i, \tag{13}$$

which are linearly independent by assumption.

Now construct a matrix $M$ with $\mathcal{I}(M) = \mathcal{I}(J_{\phi^{-1}}(\hat{\mathbf{s}}))$. By the index-set inclusion assumption, for each $l \in \mathcal{N}_i$ we have

$$\mathcal{I}\big((J_u(\mathbf{s}_t^{(l)})M)_{i,\cdot}\big) \subseteq \mathcal{I}\big((J_{\hat{u}}(\hat{\mathbf{s}}_t^{(l)}))_{i,\cdot}\big). \tag{14}$$

Thus,

$$(J_u(\mathbf{s}_t^{(l)}))_{i,\cdot} M \in \operatorname{span}\{e_j : j \in \mathcal{I}((J_{\hat{u}})_{i,\cdot})\}. \tag{15}$$

Taking linear combinations across $l \in \mathcal{N}_i$ preserves this property, so in particular,

$$M_{j,\cdot} \in \operatorname{span}\{e_k : k \in \mathcal{I}((J_{\hat{u}})_{i,\cdot})\}, \quad \forall j \in \mathcal{I}((J_u)_{i,\cdot}). \tag{16}$$

Since $J_{\phi^{-1}}(\hat{\mathbf{s}}_t)$ is invertible, there exists a permutation $\pi$ such that

$$(J_{\phi^{-1}}(\hat{\mathbf{s}}_t))_{j,\pi(j)} \neq 0, \quad \forall j \in \{1, \ldots, d_s\}. \tag{17}$$

Because $\mathcal{I}(M) = \mathcal{I}(J_{\phi^{-1}}(\hat{\mathbf{s}}_t))$, we obtain

$$M_{j,\pi(j)} \neq 0, \quad \forall j \in \mathcal{I}((J_u)_{i,\cdot}). \tag{18}$$

Combining this with Equation 16, we conclude

$$\pi(j) \in \mathcal{I}((J_{\hat{u}})_{i,\cdot}), \quad \forall j \in \mathcal{I}((J_u)_{i,\cdot}). \tag{19}$$

Equation 19 shows that each ground-truth relevant index $j \in \mathcal{I}((J_u)_{i,\cdot})$ is mapped to a distinct estimated relevant index $\pi(j) \in \mathcal{I}((J_{\hat{u}})_{i,\cdot})$. Therefore, the estimated index set must contain at least as many elements as the ground-truth one:

$$\|\mathcal{I}((J_{\hat{u}})_{i,\cdot})\| \geq \|\mathcal{I}((J_u)_{i,\cdot})\|. \tag{20}$$

This completes the proof. □

### A.6. Proof of Theorem 2

**Theorem 2.** *Consider two observationally equivalent generative processes $\mathbf{o}_t = f_t(\mathbf{s}_t)$ and $\mathbf{o}_t = \hat{f}_t(\hat{\mathbf{s}}_t)$, and assume the conditions in Proposition 2. Then, for any task $\mathbf{g}_k$ with latent index set $I_k$, with a sparsity regularization*

$$\|\mathcal{I}(J_{\hat{u}})\| \leq \|\mathcal{I}(J_u)\|,$$

*under some permutation $\pi$, the estimated task-relevant latent variables $\hat{\mathbf{s}}_{t,\pi(I_k)}$ are an invertible function $h_k$ of only the ground-truth task-relevant latent variables $\mathbf{s}_{t,I_K}$, i.e.,*

$$\hat{\mathbf{s}}_{t,\pi(I_k)} = h_k(\mathbf{s}_{t,I_K}).$$

*Proof.* The first part of the proof follows the similar strategy as Proposition 2, and we provide the full details for completeness. Since $\mathbf{o}_t = f_t(\mathbf{s}_t)$ and $\mathbf{o}_t = \hat{f}_t(\hat{\mathbf{s}}_t)$ are observationally equivalent, there exists an invertible mapping $\phi$ such that

$$\hat{\mathbf{s}}_t = \hat{f}_t^{-1} \circ f_t(\mathbf{s}_t) = \phi(\mathbf{s}_t), \tag{21}$$

with inverse $\phi^{-1}$. By the chain rule,

$$J_{\hat{u}} = J_u \, J_{\phi^{-1}}. \tag{22}$$

For each $i \in [d_g]$, consider a set $\mathcal{N}_i$ of $\|\mathcal{I}((J_u)_{i,\cdot})\|$ distinct points and the corresponding Jacobians

$$\left. \left( \frac{\partial u_i}{\partial \mathbf{s}_{t,1}}, \frac{\partial u_i}{\partial \mathbf{s}_{t,2}}, \dots, \frac{\partial u_i}{\partial \mathbf{s}_{t,d_s}} \right) \right|_{\mathbf{s}=\mathbf{s}_t^{(l)}}, \quad l \in \mathcal{N}_i. \tag{23}$$

By assumption, the vectors in Equation 23 are linearly independent.

We now construct a matrix $M$. Because the row vectors in Equation 23 are linearly independent, any row $M_{j,\cdot}$ with $j \in \mathcal{I}((J_u)_{i,\cdot})$ can be expressed as a linear combination of them. That is, there exist coefficients $\{\beta_l\}_{l \in \mathcal{N}_i}$ such that

$$M_{j,\cdot} = \sum_{l \in \mathcal{N}_i} \beta_l \, (J_u(\mathbf{s}_t^{(l)}))_{i,\cdot} \, M. \tag{24}$$

We require $M$ to satisfy two conditions: (i) for each $i \in [d_g]$, the linear combination in Equation 24 must lie in the span of the canonical basis vectors indexed by $\mathcal{I}((J_{\hat{u}})_{i,\cdot})$, i.e.,

$$\sum_{l \in \mathcal{N}_i} \beta_l \, (J_u(\mathbf{s}_t^{(l)}))_{i,\cdot} \, M \in \text{span}\{e_j : j \in \mathcal{I}((J_{\hat{u}})_{i,\cdot})\}, \tag{25}$$

and (ii) its index set matches that of $J_{\phi^{-1}}(\hat{\mathbf{s}}_t)$:

$$\mathcal{I}(M) = \mathcal{I}(J_{\phi^{-1}}(\hat{\mathbf{s}}_t)). \tag{26}$$

By the index-set inclusion assumption, for all $l \in \mathcal{N}_i$ we have

$$\mathcal{I}\big((J_u(\mathbf{s}_t^{(l)}) \, M)_{i,\cdot}\big) \subseteq \mathcal{I}\big((J_{\hat{u}}(\hat{\mathbf{s}}_t^{(l)}))_{i,\cdot}\big). \tag{27}$$

This guarantees

$$(J_u(\mathbf{s}_t^{(l)}))_{i,\cdot} \, M \in \text{span}\{e_j : j \in \mathcal{I}((J_{\hat{u}})_{i,\cdot})\}. \tag{28}$$

Taking linear combinations with the coefficients $\{\beta_l\}$, we conclude

$$\sum_{l \in \mathcal{N}_i} \beta_l \, (J_u(\mathbf{s}_t^{(l)}))_{i,\cdot} \, M \in \text{span}\{e_j : j \in \mathcal{I}((J_{\hat{u}})_{i,\cdot})\}. \tag{29}$$

Equivalently, for every $j \in \mathcal{I}((J_u)_{i,\cdot})$,

$$M_{j,\cdot} \in \text{span}\{e_k : k \in \mathcal{I}((J_{\hat{u}})_{i,\cdot})\}. \tag{30}$$

Since $J_{\phi^{-1}}(\hat{\mathbf{s}}_t)$ is invertible, its determinant is nonzero. Expanding the determinant as a sum over permutations, there must exist a permutation $\pi$ such that

$$(J_{\phi^{-1}}(\hat{\mathbf{s}}_t))_{j,\pi(j)} \neq 0, \quad \forall j \in \{1, \dots, d_s\}. \tag{31}$$

This establishes a one-to-one correspondence between the indices of $\mathbf{s}_t$ and $\hat{\mathbf{s}}_t$ through $\pi$.

In particular, for every $j \in \mathcal{I}((J_u)_{i,\cdot})$, we have

$$(J_{\phi^{-1}}(\hat{\mathbf{s}}_t))_{j,\pi(j)} \neq 0. \tag{32}$$

Because $\mathcal{I}(M) = \mathcal{I}(J_{\phi^{-1}}(\hat{\mathbf{s}}_t))$, this implies

$$M_{j,\pi(j)} \neq 0, \quad \forall j \in \mathcal{I}((J_u)_{i,\cdot}). \tag{33}$$

Combining this with Equation 30, it follows that

$$\pi(j) \in \mathcal{I}((J_{\hat{u}})_{i,\cdot}), \quad \forall j \in \mathcal{I}((J_u)_{i,\cdot}). \tag{34}$$

Therefore, every nonzero entry of $J_u$ has a corresponding nonzero entry of $J_{\hat{u}}$ at the permuted column index:

$$(J_u)_{i,j} \neq 0 \implies (J_{\hat{u}})_{i,\pi(j)} \neq 0. \tag{35}$$

Finally, with the sparsity regularization $\|J_{\hat{u}}\|_0 \leq \|J_u\|_0$, this implication strengthens to an equivalence:

$$(J_u)_{i,j} \neq 0 \iff (J_{\hat{u}})_{i,\pi(j)} \neq 0. \tag{36}$$

For $c \in I_k$, we have $c \in \mathcal{I}((J_u)_{k,\cdot})$. Hence, by Equation 30,

$$M_{c,\cdot} \in \mathrm{span}\{e_{k'} : k' \in \mathcal{I}((J_{\hat{u}})_{k,\cdot})\}. \tag{37}$$

Suppose, for contradiction, that $M_{c,\pi(r)} \neq 0$ for some $r \in I \setminus I_k$. Then $\pi(r)$ belongs to the index set on the right-hand side of Equation 37.

By Equation 36, this implies that $r \in \mathcal{I}((J_u)_{k,\cdot})$, i.e. $r \in I_k$, contradicting $r \in I \setminus I_k$. Therefore, $M_{c,\pi(r)} = 0$, which together with $\mathcal{I}(M) = \mathcal{I}(J_{\phi^{-1}}(\hat{\mathbf{s}}_t))$ yields

$$\frac{\partial \mathbf{s}_{t,c}}{\partial \hat{\mathbf{s}}_{t,\pi(r)}} = 0, \quad \forall c \in I_k, \ r \in I \setminus I_k. \tag{38}$$

Since $\phi$ is invertible, there exists an invertible mapping between $\mathbf{s}_{t,c}$ and $\hat{\mathbf{s}}_{t,\pi(c)}$, and $\mathbf{s}_{t,c}$ depends only on $\hat{\mathbf{s}}_{t,\pi(c)}$. Moreover, because $r \in I \setminus I_k$ and $c \in I_k$, $\mathbf{s}_{t,r}$ is independent of $\mathbf{s}_{t,c}$. Hence, $\mathbf{s}_{t,r}$ does not depend on $\hat{\mathbf{s}}_{t,\pi(c)}$, in the sense that their mutual information is zero. Thus, we further have

$$\frac{\partial \mathbf{s}_{t,r}}{\partial \hat{\mathbf{s}}_{t,\pi(c)}} = 0, \quad \forall c \in I_k, \ r \in I \setminus I_k. \tag{39}$$

Given the invertibility of the mapping between $\mathbf{s}_t$ and $\hat{\mathbf{s}}_t$, the inverse of both Equations 38 and 39 also holds. Thus, the only dependencies remain within the estimated and ground-truth task-relevant parts, completing the proof. $\square$

# B. Supplementary Discussions

## B.1. Further Comparison with Related Works

**Learning Temporal Task Structure.** For our temporal task structure results in Section 3, the most relevant prior work is Qiu et al. (2024), which also models tasks as colliders and seeks to recover their structure in an unsupervised manner. However, the problem we considered is *fundamentally different* as follows:

- **Theoretical Foundation vs. Heuristic Decomposition.** The most essential distinction lies in identifiability. As noted in Section 3, SelTask relies on sequential non-negative matrix factorization, which is a purely heuristic decomposition without any identifiability guarantees for recovering the true temporal task structure. Identifiability is critical because it sets the ultimate limit of any model and provides the guarantee of recovering the ground-truth representation. Our work fills this theoretical gap by providing the first general nonparametric identifiability guarantee for this problem, ensuring the recovered structure is faithful to the underlying generative process.

- **More General Problem Setting.** In addition to providing theoretical guarantees, our approach generalizes the previous heuristic methods (including SelTask) in several crucial ways. First, our theory accommodates tasks that may appear, disappear, and interleave arbitrarily over time, moving beyond the assumption of sequential completion typical of decomposition-based methods like SelTask. Second, we do not require strict temporal dependence, and our framework accounts for sequences that may contain an arbitrary number of disconnected boundaries or even i.i.d. settings. SelTask does not support such temporal disconnections. Lastly, the data-generating process is fully nonparametric, allowing for complex nonlinear functions and arbitrary distributions, without relying on auxiliary information or distributional constraints.

**Learning Task Relevant Representation.** Many previous works study the identification of latent variables that generate observational data, but they do not address our goal of recovering a task-relevant representation. On the technical side, some works also adopt a structural view of the hidden generative process. A key example is Zheng et al. (2022), which establishes identifiability results for nonlinear ICA under structural conditions. To avoid confusion, we clarify the differences between their setting and ours as follows:

- **Different settings.** Zheng et al. (2022) considers the identifiability of nonlinear ICA in the IID setting, while we consider the general temporal settings.

- **Different goals.** Zheng et al. (2022) aims to recover all individual latent variables, while we aim to recover the temporal task structure, and task-relevant variables as a group.

- **Different assumptions.** Zheng et al. (2022) assumes structural sparsity, i.e., a specific graphical criterion on the underlying structure between latent and observed variables; while we do not impose these constraints on the data generative process. Moreover, Zheng et al. (2022) assumes latent independence while we do not.

- **Different proof strategies.** Since the considered problems and techniques are fundamentally different, naturally, the proof strategies differ. If we treat task variables as observed variables, the proof ideas align up to the point where we recover the support of the Jacobian up to permutation. However, this yields $J_{\hat{u}}(\hat{s}) = D_1 J_u(s) D_2 P$, which actually says *nothing* about the identifiability of latent variables. The latent components can still be mixed arbitrarily, including across the sets associated with different tasks, let alone within each set.

  Previous works relying on Structural Sparsity use that assumption to go further and achieve element-wise identifiability, obtaining $J_{\hat{u}}(\hat{s}) = J_u(s) D P$. Our setting does not require that level of identifiability. The central question for us is more modest but crucial: for two task variables $u_1$ and $u_2$, how do we ensure that estimated latents associated with $u_1$ do not mix with ground-truth latents associated with $u_2$? This separation cannot be guaranteed by any existing proof logic. Our result provides exactly that guarantee without imposing structural sparsity.

### B.2. Detailed Discussion on Main Conditions

For identifying task-relevant representations, our main condition is that in Proposition 2. The assumptions are intended to ensure that the Jacobian carries enough variation to span its support, thereby capturing the underlying dependencies between latent state variables and task variables in the nonlinear setting. While these assumptions may appear technical at first, they are usually quite mild in practice. The span condition requires that across a small number of samples, the Jacobian vectors for each task variable $u_i$ span the relevant support. Intuitively, this rules out degenerate situations where all data points come from an extremely narrow subpopulation that fails to exhibit the necessary variation. In typical settings with smooth mappings and a latent state distribution that has a continuous density, Jacobian evaluations at independently drawn samples form continuous random vectors that are in general position with probability one. This means that only $|\mathcal{I}(J_u(\mathbf{s}t)i, \cdot)|$ random samples are usually enough to span the support, and this number corresponds to how many latent state variables are relevant to $u_i$, which is usually much smaller than the full sample size. As a result, the condition is typically satisfied with very few datapoints. The index set inclusion assumption is also mild. Since

$$J_u(\mathbf{s}_t) M'(\mathbf{s}_t, \hat{\mathbf{s}}_t) = J_{\hat{u}}(\hat{\mathbf{s}}_t), \tag{40}$$

and $M$ shares the same nonzero index pattern as $M'$, the row $(J_u(\mathbf{s}_t) M)_{i,\cdot}$ already lies inside the support of $(J_{\hat{u}}(\hat{\mathbf{s}}_t))_{i,\cdot}$. While special value combinations could in principle cause the supports to differ at particular points, the condition only requires the existence of one such Jacobian in the relevant space, which is almost always satisfied in practice.

## C. Supplementary Experimental Setups

In this section, we include further details of the experimental setups not fully elaborated in the main text because of space constraints. In summary, our practical CI implementation is aligned with existing high-dimensional practice. When the variables are high-dimensional, it is routine to first map them into a lower-dimensional representation and then estimate conditional mutual information in that reduced space. This representation step is widely used in CMI based CI testing to avoid the curse of dimensionality. It is also common to approximate CMI with scalable variational bounds. Neural and variational CMI estimators (Belghazi et al., 2018; Molavipour et al., 2020; Mondal et al., 2020) and contrastive objectives such as InfoNCE (Oord et al., 2018; Sordoni et al., 2021) follow exactly this paradigm and have been used extensively in high dimensional dependence estimation. Our method adopts the same blueprint.

**CMI Surrogate for the CI Test.** In high-dimensional settings, when direct conditional independence (CI) testing is computationally infeasible, it is standard to use approximations such as conditional mutual information (CMI). Below we provide additional details on this surrogate.

For each pair $(\mathbf{S}_k, \mathbf{S}_v)$ and task $\mathbf{g}_i$, we replace the CI test

$$H_0: \ \mathbf{s}_{kL} \perp \mathbf{s}_{vL} \ \big| \ \mathbf{Z}_{\mathrm{band}}(k, v, i), \tag{41}$$

with an estimate of the conditional mutual information

$$I\big(\mathbf{s}_{kL}; \mathbf{s}_{vL} \mid \mathbf{Z}_{\mathrm{band}}(k, v, i)\big) = \mathbb{E}\left[\log \frac{p(\mathbf{s}_{kL}, \mathbf{s}_{vL} \mid \mathbf{Z}_{\mathrm{band}})}{p(\mathbf{s}_{kL} \mid \mathbf{Z}_{\mathrm{band}}) \, p(\mathbf{s}_{vL} \mid \mathbf{Z}_{\mathrm{band}})}\right]. \tag{42}$$

Direct estimation with $\mathbf{Z}_{\mathrm{band}}$ can be high dimensional. We therefore learn a task-conditioned representation $\mathbf{c}_i = h_\phi\big(\mathbf{Z}_{\mathrm{band}}(k, v, i)\big)$ and instead test with $I(\mathbf{s}_{kL}; \mathbf{s}_{vL} \mid \mathbf{c}_i)$. If $h_\phi$ is conditionally sufficient for $\mathbf{Z}_{\mathrm{band}}$ with respect to $\{\mathbf{s}_{kL}, \mathbf{s}_{vL}\}$, then

$$I(\mathbf{s}_{kL}; \mathbf{s}_{vL} \mid \mathbf{Z}_{\mathrm{band}}) = I(\mathbf{s}_{kL}; \mathbf{s}_{vL} \mid \mathbf{c}_i).$$

We estimate a variational lower bound on $I(\mathbf{s}_{kL}; \mathbf{s}_{vL} \mid \mathbf{c}_i)$ using a conditional InfoNCE objective. Let $f_\theta(\mathbf{s}_{kL}, \mathbf{s}_{vL}, \mathbf{c}_i)$ be a critic. For each positive pair $(\mathbf{s}_{kL}, \mathbf{s}_{vL}, \mathbf{c}_i)$, draw $K$ negatives $\{\tilde{\mathbf{s}}_{vL}^{(j)}\}_{j=1}^K$ by shuffling $\mathbf{s}_{vL}$ within mini-batches that share $\mathbf{c}_i$ (or within nearest neighbors of $\mathbf{c}_i$). Optimize

$$\mathcal{L}_{\mathrm{cNCE}}(\theta, \phi) = \mathbb{E}\left[\log \frac{\exp f_\theta(\mathbf{s}_{kL}, \mathbf{s}_{vL}, \mathbf{c}_i)}{\exp f_\theta(\mathbf{s}_{kL}, \mathbf{s}_{vL}, \mathbf{c}_i) + \sum_{j=1}^K \exp f_\theta(\mathbf{s}_{kL}, \tilde{\mathbf{s}}_{vL}^{(j)}, \mathbf{c}_i)}\right], \tag{43}$$

which lower bounds $I(\mathbf{s}_{kL}; \mathbf{s}_{vL} \mid \mathbf{c}_i)$ up to a constant. After training a single task-conditioned critic across all $(k, v, i)$, define

$$\widehat{I}_{\mathrm{cNCE}}(k, v, i) = \mathbb{E}\left[\log \frac{\exp f_\theta(\mathbf{s}_{kL}, \mathbf{s}_{vL}, \mathbf{c}_i)}{\frac{1}{K} \sum_{j=1}^K \exp f_\theta(\mathbf{s}_{kL}, \tilde{\mathbf{s}}_{vL}^{(j)}, \mathbf{c}_i)}\right]. \tag{44}$$

We reject $H_0$ for $(k, v, i)$ if $\widehat{I}_{\mathrm{cNCE}}(k, v, i)$ exceeds a permutation threshold obtained by re-sampling $\{\tilde{\mathbf{s}}_{vL}^{(j)}\}$ within $\mathbf{c}_i$ buckets.

**Additional Details of SportsHHI.** SportsHHI contains $11,398$ video sequences, partitioned into short clips of 5 frames each, with $55,631$ annotated pairwise interaction instances. The HHID task labels the interaction for each pair of human actors in a video; interactions often occupy short temporal windows embedded in long sequences, so a single sequence typically contains multiple, possibly overlapping interactions. This results in complex temporal patterns, with flexible interactions across multiple actors.

In our implementation of Algorithm 1 on SportsHHI, we set the number of latent state variables to be the same as the number of humans at frame $t$. For all baselines, we use a pretrained CLIP encoder (Radford et al., 2021) with a ResNet-50 backbone to get the observed RGB features $\mathbf{o}$. To handle temporal dynamics, an MLP parameterizes transitions $\mathbf{s}_{t-1} \to \mathbf{s}_t$, while conditional mutual information (CMI) is estimated on latent trajectories as a surrogate for conditional independence testing. To ensure fairness, all baselines employ a ResNet-50 backbone for RGB feature extraction, consistent with prior work.

**Downstream Benefit.** We evaluate our method on the Meta-World benchmark (Yu et al., 2020) by constructing an interleaved offline dataset from the *door-open/close; drawer-open* tasks. Both tasks involve a 7-DoF robotic arm manipulating the same door but with opposite goals, making them an ideal testbed for multi-task interference. We first train task-specific expert policies using SAC until reaching $60\%$ success rate, then collect $\sim 300$ successful and $\sim 300$ mixed-quality trajectories for each task. To create interleaved data, we segment trajectories into 30–60 step skill chunks (e.g., reaching, grasping, rotating). With probability $p = 0.8$, we randomly splice open- and close-task segments into a single trajectory, inserting short transition phases to ensure physical continuity. This results in $\sim 2.4$k interleaved trajectories, with on average 2.1 task switches per trajectory. We provide only weak or noisy task labels derived from the door angle change, simulating realistic partially labeled data. We build upon the Active Fine-Tuning (AMF) framework (Bagatella et al., 2025). Specifically, the agent learns a policy over identified tasks using their representation $\mathbf{g}$, which replaces the task embedding $\mu_c$ in AMF. This enables the agent to actively select tasks that improve generalization. To evaluate this, we train on three tasks—*door-open*, *door-close*, and *drawer-open*, and test generalization to the new task *drawer-close* with only $10^4$ samples.

*Table 2.* Runtime (expected value) under varying number of time steps.

| $T$ | 8 | 10 | 12 | 14 | 16 | 18 | 20 |
|---|---|---|---|---|---|---|---|
| Ours | 0.01 | 0.01 | 0.01 | 0.01 | 0.01 | 0.01 | 0.02 |
| CCA | 0.01 | 0.01 | 0.02 | 0.02 | 0.02 | 0.03 | 0.04 |
| Group Lasso | 11.2 | 26.8 | 34.8 | 36.3 | 58.3 | 59.7 | 82.3 |
| SelTask | 0.69 | 1.12 | 1.40 | 1.33 | 1.82 | 2.11 | 2.40 |

| With sparsity | Without sparsity |
|---|---|

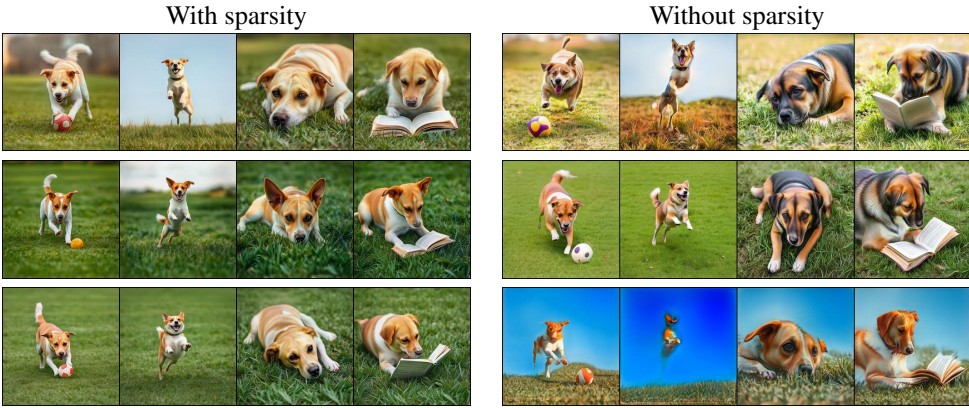

*Figure 7.* Comparison of controllable generation for a task of "a dog playing ball, jumping high, lying on the grass, and reading a book". With sparsity, the model learns task-relevant latent representations and modifies only the intended concepts for each instruction. Without sparsity, the representation entangles irrelevant factors, causing unintended changes and reduced precision.

## D. Supplementary Experimental Results

**Runtime Analysis.** we have conducted additional runtime analysis on seven datasets with four methods. To test our algorithm in the most computationally-heavy case, we set the segment lengths to the minimal value of 2. Following the same setting in the manuscript, we vary the number of time steps from 8 to 20, and set the number of tasks $M = T/5$. The results (seconds) are as Table 2.

**Additional Results on Learning Temporal Task Structure.** To further study the ability of different models to recover temporal task structure, we evaluated several additional approaches on the task structure prediction benchmark. The goal is to compare with more standard video models that do not target identifiability. We further included two new video models, Slowfast (Feichtenhofer et al., 2019) and VitB (Tong et al., 2022). As shown in Table 3, our method achieves the best performance, which further validates that a principled structure learning approach yields the most reliable recovery of temporal task structure. Leap outperforms the Base model, which highlights the benefit of identifiable representations for structure learning. However, when observations are modeled more appropriately, this advantage becomes less pronounced, as seen in the similar performance of Slowfast and VitB.

*Table 3.* Additional results on SportsHHI

| Method | mAP |
|---|---|
| Ours | $0.25 \pm 0.08$ |
| Leap | $0.12 \pm 0.05$ |
| Base | $0.09 \pm 0.01$ |
| Slowfast | $0.11 \pm 0.02$ |
| VitB | $0.12 \pm 0.03$ |

**Additional Results on Controllable Generation.** Moreover, we conduct additional experiments to evaluate the benefit of recovering task relevant representations for controllable generation. We consider the task of "a dog playing ball, jumping high, lying on the grass, and reading a book". The empirical setup follows that of Figure 6. From Figure 7, it becomes evident that precise control requires learning task-relevant representations with sparsity regularization. Without it, the learned representation absorbs irrelevant hidden factors and introduces unwanted changes in the generated outputs. For instance, the dog may change into a different one, its shape may become unnatural, and sometimes irrelevant backgrounds dominate the generation.

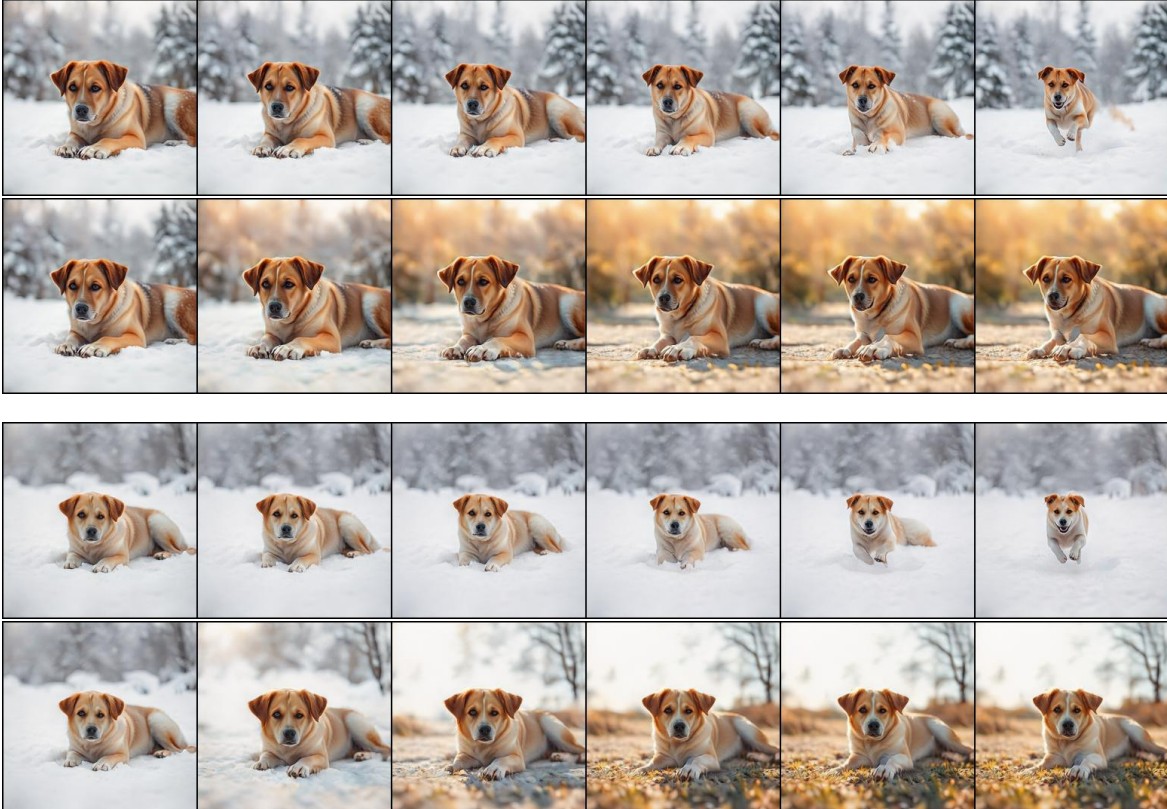

Identified latents corresponding to "running" and "season" are successfully identified. Even in complex settings where attributes are not clearly separable visually, the method recovers meaningful representations. This also shows how the identified latents vary across tasks in an interpretable way.

