# OpenReview forum: "From Generalist to Specialist Representation"
_ICML.cc/2026/Conference — ICML 2026 regular_

### Official Review · Reviewer_9eTs · 2026-03-12

**Soundness:** 3
**Presentation:** 2
**Significance:** 3
**Originality:** 3
**Overall Recommendation:** 3
**Confidence:** 3

**Summary:**

This paper establishes a two-level nonparametric identifiability framework for recovering task-relevant latent variables from observations: first identifying temporal task structure across time steps, then disentangling task-relevant components of the latent state within each step via sparsity regularization.

**Compliance With Llm Reviewing Policy:**

Affirmed.

**Key Questions For Authors:**

The theoretical framework critically relies on actions $a_t$​, yet no main experiment evaluates the method in settings with real-world physical interaction. How does the method perform in domains such as robotic manipulation where actions have direct consequences in the world?

**Limitations:**

The empirical validation does not fully support the nonparametric theoretical claims, leaving open the question of how the method behaves in genuinely nonlinear and physically interactive settings.

I am happy to raise my rating if these concerns on the empirical study are addressed.

**Strengths And Weaknesses:**

Strengths:
1. Nonparametric identifiability results of this generality are limited in the literature, representing a meaningful advance over work that relies on linear assumptions, intervention targets, or distributional constraints.
2. The modeling of tasks as colliders is particularly elegant. A single conditioning set simultaneously blocks temporal paths through non-collider states and activates task-mediated paths through the collider mechanism, and grounding this into a rigorous identifiability guarantee is the strongest technical contribution of the paper.

Weaknesses:
1. The empirical validation does not fully support the nonparametric claims: synthetic experiments are conducted exclusively on linear Gaussian data with Fisher's z-test, leaving the behavior of the method under genuine nonlinearity and non-Gaussian settings unclear. While real-world results show clear improvement over baselines, the high variance raises questions about stability.
2. Qualitative vision experiments rely on synthetically generated images with visually well-separated task attributes, making it difficult to assess whether the method generalizes to more realistic and challenging scenarios.
3. The paper lacks intuitive visualizations that demonstrate what the method is actually recovering — for instance, how identified task-relevant latents vary across tasks, or how recovered temporal structure aligns with ground truth in an interpretable way.
4. The theoretical exposition is at times difficult to follow, which hinders accessibility.

---

> ### Author Rebuttal · Authors · 2026-03-31
>
> We sincerely appreciate your insightful feedback. In light of it, we add **new experiments, analyses, and discussions** as follows:
>
> ---
>
> **Q1.** Experiments in genuine nonlinear and non-Gaussian settings.
>
> **A1.** Thanks so much for your suggestion. Since the synthetic experiments for task-relevant representation (Fig. 5) already use a nonparametric data-generating process (L400, left), we add new nonlinear and non-Gaussian experiments for temporal task-structure learning.
>
> Specifically, we follow Fig. 3, but generate observations using an MLP with leaky ReLU and non-Gaussian noise, and adopt a kernel-based CI test. MCCs are:
>
> |Method|T=8|T=10|T=12|T=14|T=16|T=18|T=20|
> |-|-|-|-|-|-|-|-|
> |CCA|0.25±0.13|0.17±0.11|0.24±0.08|0.27±0.10|0.24±0.15|0.31±0.08|0.26±0.10|
> |GroupLasso|0.44±0.13|0.28±0.13|0.35±0.06|0.32±0.11|0.28±0.11|0.19±0.15|0.21±0.17|
> |SelTask|0.42±0.16|0.46±0.18|0.30±0.12|0.32±0.16|0.25±0.10|0.41±0.17|0.44±0.16|
> |Ours|0.81±0.10|0.82±0.13|0.69±0.14|0.60±0.13|0.60±0.15|0.51±0.12|0.55±0.18|
>
> |Method|M=2|M=4|M=6|M=8|M=10|
> |-|-|-|-|-|-|
> |CCA|0.17±0.12|0.18±0.11|0.15±0.10|0.20±0.09|0.21±0.10|
> |GroupLasso|0.11±0.08|0.15±0.09|0.17±0.08|0.18±0.07|0.18±0.07|
> |SelTask|0.18±0.14|0.18±0.13|0.22±0.12|0.21±0.11|0.23±0.10|
> |Ours|0.35±0.13|0.40±0.16|0.42±0.15|0.41±0.14|0.43±0.13|
>
> Our method consistently outperforms baselines across all nonlinear and non-Gaussian settings.
>
> For task-relevant representation, we further add a nonlinear baseline (iVAE [1]). We evaluate the gap between the $R^2$ of the task-relevant and irrelevant parts across different dimensionalities. If task-relevant representations are identifiable, this gap should be large; otherwise, it should be small, since the model cannot reliably distinguish the two (cf. Fig. 5).
>
> |Method|10|20|30|40|50|60|70|80|90|100|
> |-|-|-|-|-|-|-|-|-|-|-|
> |iVAE|0.10±0.04|0.06±0.08|0.07±0.11|0.05±0.04|0.06±0.07|0.10±0.08|0.09±0.02|0.10±0.05|0.03±0.04|0.13±0.11|
> |Ours|0.63±0.07|0.66±0.06|0.65±0.04|0.66±0.05|0.72±0.08|0.66±0.07|0.65±0.04|0.63±0.09|0.70±0.09|0.65±0.07|
>
> We observe that iVAE fails to separate relevant from irrelevant ones, while our method consistently achieves a large gap. This is expected, as iVAE relies on sufficient auxiliary information, which is not present in our setting.
>
>
> [1] Variational Autoencoders and Nonlinear ICA: A Unifying Framework
>
> ---
>
> **Q2.** Visual experiments on more challenging scenarios & how identified latents vary across tasks.
>
> **A2.** Thanks so much for the suggestions. Accordingly, we have added new visual results at this [anonymous link](https://anonymous.4open.science/r/visual_results-DB77/new_visual_results.png).
>
> The results show that even when attributes are not cleanly separable visually, such as multiple factors jointly affecting global appearance across seasons, the method still recovers meaningful latent representations. Moreover, the identified latents vary in an intuitive and interpretable way across tasks, clearly capturing distinct factors such as motion (e.g., running) and environmental changes (e.g., season).
>
> ---
>
> **Q3.** More intuitive theoretical exposition.
>
> **A3.** Thank you for the helpful feedback. We have added more intuitive explanations and examples throughout the paper to clarify the main ideas. For instance, we expanded the discussion of Proposition 2 and made similar revisions to better connect the theory with intuition:
>
> > “Proposition 2 implies that foundation models are sufficiently expressive to include all task-relevant latents, but may also capture task-irrelevant components, leading to unnecessarily large representations. This motivates our next result to isolate the task-relevant part.”
>
> ---
>
> **Q4.** Experiments in physically interactive settings
>
> **A4.** Thank you for the suggestion. In light of it, we evaluate our method on the *Meta-World* [2], a widely-used benchmark for the physical interactions of robots.
>
> We construct an interleaved offline dataset from multiple tasks. Each task involves a 7-DoF robotic arm manipulating the same object but with opposite goals. We collect expert trajectories with Soft Actor Critic [3] and construct interleaved data by segmenting skills and splicing across tasks with transition phases. We extend the Active Fine-Tuning [4] with our task discovery algorithm, and the latent states are obtained via task-latent sparsity regularization on PWM [5]. We train on interleaved tasks (*door-open*, *door-close*, and *drawer-open*) and test generalization to the new task (*drawer-close*) with only $10^4$ samples. The success rates show that our theory enables the model to learn and generalize more efficiently.
>
> |Ours|CCA|GroupLasso|SelTask|
> |-|-|-|-|
> |0.75|0.50|0.46|0.70|
>
> [2] Meta-World: A Benchmark and Evaluation for Multi-Task and Meta Reinforcement Learning
>
> [3] Soft Actor-Critic: Off-Policy Maximum Entropy Deep Reinforcement Learning with a
> Stochastic Actor
>
> [4] Active Fine-Tuning of Multi-Task Policies
>
> [5] PWM: Policy Learning with Multi-Task World Models

---

### Official Review · Reviewer_jozk · 2026-03-13

**Soundness:** 3
**Presentation:** 2
**Significance:** 2
**Originality:** 2
**Overall Recommendation:** 5
**Confidence:** 3

**Summary:**

This paper provides the general nonparametric identifiability guarantees for learning task-relevant latent representations from observations, requiring no parametric assumptions, interventions, or side information. The approach works hierarchically: it first recovers the temporal task structure even under arbitrary disconnections and interleaving, then uses sparsity regularization to provably separate task-relevant latent variables from nuisance factors within each time step, with experiments on both synthetic and real-world datasets validating the theoretical results. These findings lay a theoretical foundation for principled specialization of foundation models, showing that simple fine-tuning with sparsity constraints is sufficient to extract minimal, faithful, task-specific representations from general-purpose models.

**Compliance With Llm Reviewing Policy:**

Affirmed.

**Final Justification:**

Authors add more experimental results during the rebuttal and addressed my question so I decided to increase my score.

**Key Questions For Authors:**

How robust is the approach to different sparsity regularization coefficients, and can the appropriate level of regularization be determined in practice without knowing the true task-relevant variables?

**Limitations:**

yes

**Strengths And Weaknesses:**

Soundness:
The paper's claims are technically correct and generally well-supported and explained. Both synthetic and real-world dataset experiments are conducted and discussed. Authors also address the limitations. The proof structure is clean and builds naturally from task structure to latent disentanglement, with reasonable assumptions. Experiments support the theory on both synthetic and real-world settings, though broader empirical evaluation across more domains would further solidify the claims.

Presentation:
Paper is really well written and easy to follow. The theoretical sections are clearly structured with intuitive proof sketches and helpful visual examples (i.e. Figure 2), making the hierarchical framework accessible. Yet, the experimental results section is not presented in detail unlike the theoretical sections.

Significance:
The paper tackles a core question in representation learning — whether task-relevant latents can be provably recovered — which is broadly relevant to the foundation model era where specialization through fine-tuning is standard practice. The practical impact remains bounded by the asymptotic setting, but the framework opens promising directions for idetifiability-driven architecture design and more principled approaches to model specialization.

Originality:
The key contribution is extending identifiability guarantees to a fully nonparametric setting, removing the need for interventions, parametric forms, or auxiliary information. The individual tools, such as sparsity regularization and conditional independence testing, are well-established, but achieving identifiability guarantees under a fully nonparametric setting where prior work required stronger assumptions is simple yet novel.

---

> ### Author Rebuttal · Authors · 2026-03-31
>
> We are very grateful for the constructive feedback, and add **new experiments, analyses, and discussions** as follows:
>
> ---
>
> **Q1.** Experiments support the theory in both synthetic and real-world settings, though broader empirical evaluation and more details would further solidify the claims.
>
> **A1.** Thanks for the constructive suggestion. We have further broadened the empirical evaluation with additional experiments.
>
> First, we have conducted **new nonlinear synthetic experiments**. Specifically, we follow the same setup as in Fig. 3, but generate observations using an MLP with leaky ReLU, and adopt a kernel-based conditional independence test as a fully nonparametric CI estimator. We report MCC as follows, and observe that our method consistently outpeforms baselines.
>
> |Method|T=8|T=10|T=12|T=14|T=16|T=18|T=20|
> |-|-|-|-|-|-|-|-|
> |CCA|0.25±0.13|0.17±0.11|0.24±0.08|0.27±0.10|0.24±0.15|0.31±0.08|0.26±0.10|
> |GroupLasso|0.44±0.13|0.28±0.13|0.35±0.06|0.32±0.11|0.28±0.11|0.19±0.15|0.21±0.17|
> |SelTask|0.42±0.16|0.46±0.18|0.30±0.12|0.32±0.16|0.25±0.10|0.41±0.17|0.44±0.16|
> |Ours|0.81±0.10|0.82±0.13|0.69±0.14|0.60±0.13|0.60±0.15|0.51±0.12|0.55±0.18|
>
>
> |Method|M=2|M=4|M=6|M=8|M=10|
> |-|-|-|-|-|-|
> |CCA|0.17±0.12|0.18±0.11|0.15±0.10|0.20±0.09|0.21±0.10|
> |GroupLasso|0.11±0.08|0.15±0.09|0.17±0.08|0.18±0.07|0.18±0.07|
> |SelTask|0.18±0.14|0.18±0.13|0.22±0.12|0.21±0.11|0.23±0.10|
> |Ours|0.35±0.13|0.40±0.16|0.42±0.15|0.41±0.14|0.43±0.13|
>
>
> Second, we include a **sensitivity analysis** on the CI test threshold. We vary the significance level with $T=10$, $M=2$, averaged over five runs:
>
>
> |Threshold|0.005|0.01|0.05|0.1|
> |-|-|-|-|-|
> |MCC|0.75±0.13|0.82±0.13|0.76±0.19|0.65±0.22|
> |Accuracy|0.81±0.05|0.85±0.10|0.79±0.09|0.70±0.18|
>
> Performance is stable across thresholds, with moderate values best; large thresholds introduce false positives.
>
>
> Third, we add a **comparison with nonlinear ICA (iVAE)**. Specifically, we evaluate the gap between the $R^2$ of the task-relevant and irrelevant components across different dimensionalities. If task-relevant representations are identifiable, this gap should be large; otherwise, it should be small, since the model cannot reliably distinguish the two (as also illustrated in Fig. 5).
>
> |Method|10|20|30|40|50|60|70|80|90|100|
> |-|-|-|-|-|-|-|-|-|-|-|
> |iVAE|0.10±0.04|0.06±0.08|0.07±0.11|0.05±0.04|0.06±0.07|0.10±0.08|0.09±0.02|0.10±0.05|0.03±0.04|0.13±0.11|
> |Ours|0.63±0.07|0.66±0.06|0.65±0.04|0.66±0.05|0.72±0.08|0.66±0.07|0.65±0.04|0.63±0.09|0.70±0.09|0.65±0.07|
>
> We observe that iVAE fails to separate relevant from irrelevant components, while our method consistently achieves a large gap. This is expected, as iVAE relies on sufficient auxiliary information, which is not available in our setting.
>
> Finally, we include a **scalability analysis** of CI testing. To test the most computationally demanding case, we set $L=2$, vary $T\in[8,20]$, and set $M=T/5$. Results (seconds):
>
>
> |T|8|10|12|14|16|18|20|
> |-|-|-|-|-|-|-|-|
> |CCA|0.013±0.009|0.013±0.000|0.016±0.000|0.018±0.000|0.024±0.000|0.027±0.000|0.035±0.000|
> |GroupLasso|11.220±3.527|26.769±5.488|34.846±4.153|36.304±3.110|58.315±0.872|59.728±1.453|82.287±0.154|
> |SelTask|0.692±0.040|1.123±0.002|1.400±0.114|1.334±0.117|1.823±0.041|2.114±0.009|2.398±0.011|
> |Ours|0.008±0.007|0.006±0.000|0.008±0.000|0.010±0.000|0.013±0.000|0.014±0.000|0.018±0.001|
>
> From the results, our method is consistently faster than all baselines. The runtime increases smoothly with $T$ and remains very small in practice, suggesting good scalability in realistic regimes.
>
> Thanks again for these constructive suggestions. We have included all the new experiments and details in the updated manuscripts, which further widen the range of the empirical evaluation.
>
> ---
>
> **Q2.** More results and discussion regarding the sparsity regularization coefficient
>
> **A2.** Thanks so much for the constructive suggestion.  In light of it, we have added additional results on the regularization coefficient. Following Fig. 5, we evaluate the gap between the $R^2$ of task-relevant and irrelevant components ($d=10$), where a larger gap indicates better disentanglement:
>
>
> |Coefficient|1e-3|2e-3|1e-2|
> |-|-|-|-|
> |Gap|0.56±0.10|0.63±0.07|0.62±0.11|
>
> Performance is stable across coefficients, with moderate regularization performing best.
>
> For selection without ground-truth latents, we adopt a consistency-based criterion: choose the coefficient that yields stable representations across runs (e.g., different random seeds/initializations). Consistently recovered latents are more likely to reflect true task-relevant structure. This is closely related to stability-based model selection, and we have highlighted this in the updated manuscript. Thank you again for your very helpful suggestion.

---

> > ### Author Rebuttal · Reviewer_jozk · 2026-04-03
> >
> > Thank you for your response and new experiments with additional analyses. I am raising my score.

---

> > > ### Author Response · Authors · 2026-04-04
> > >
> > > Dear Reviewer jozk,
> > >
> > > Thank you so much for your encouragement and constructive feedback, we truly appreciate it. Just a quick note to follow up regarding the score update, as it does not yet seem to be reflected on our side.
> > >
> > > Many thanks,
> > >
> > > Authors of Submission 5542

---

### Official Review · Reviewer_Voe2 · 2026-03-13

**Soundness:** 3
**Presentation:** 3
**Significance:** 3
**Originality:** 4
**Overall Recommendation:** 5
**Confidence:** 4

**Summary:**

This paper develops a theoretical framework for identifying task-relevant latent representations from observations $o_t = f_t(s_t)$ in a multi-task setting where each task depends on a different subset of latent variables. The contribution has two main results. First, Theorem 1 shows that the temporal task structure (which tasks are active at which time steps and which latent variables each task depends on) is identifiable via conditional independence tests using "band conditioning sets" $Z_{\\text{band}}(k, v, i)$, exploiting the collider structure of tasks in the underlying SCM. This requires Markov and Faithfulness assumptions and a minimum segment length $L \\geq 2$. Second, Theorem 2 shows that task-relevant latent representations are identifiable (up to invertible functions of the true task-relevant latents) via $\\ell_1$ sparsity regularization applied within each time step. The framework is fully nonparametric, imposing no parametric constraints on the mixing functions $f_t$. Proposition 2 establishes that generalist models learn a superset of each task's relevant variables, motivating the specialist pruning step. Experiments cover synthetic linear Gaussian SCMs (quantitative, with accuracy and MCC metrics), SportsHHI video data (qualitative task structure discovery), and cat images generated by Flux (qualitative demonstration of sparsity-based disentanglement).

**Compliance With Llm Reviewing Policy:**

Affirmed.

**Final Justification:**

Solid paper, I'm happy with the rebuttal and retain Accept.

**Key Questions For Authors:**

### Questions for the Authors

1. What happens when $L = 1$, i.e., when some tasks are active for only a single time step? Does the method fail entirely, or is there a graceful degradation in identifiability? Can we test this empirically?

2. Can you provide a nonlinear synthetic experiment (e.g., with a neural network mixing function and known ground-truth latents) that validates Theorem 2 in the nonparametric setting? The linear Gaussian experiments, while clean, do not test the most novel aspect of the theory.

3. How sensitive is the method to the choice of CI test threshold (or significance level)? In the synthetic experiments, how does accuracy and MCC change as the threshold varies?

4. How does the method scale with the number of tasks and latent dimensions? What is the computational cost of the CI testing phase (Algorithm 1) as a function of $K$ (tasks), $T$ (time steps), and $d$ (latent dimension)?

### Missing References

- Klindt et al. (2021), "Towards Nonlinear Disentanglement in Natural Data with Temporal Sparse Coding" (SlowVAE), which also uses temporal structure for identifiability.
- Uhler et al. (2013), "Geometry of the Faithfulness Assumption," which provides a geometric perspective on when and why faithfulness can fail, directly relevant to the paper's reliance on this assumption.

### Suggestions

The strongest improvement would be a nonlinear synthetic experiment that validates the full pipeline (structure recovery + representation identification) with ground-truth latents and a nonlinear mixing function. This would bridge the gap between the nonparametric theory and the linear Gaussian experiments.

A sensitivity analysis over the CI test threshold and the segment length $L$ would help practitioners understand when the method is reliable. Similarly, a robustness experiment testing performance under approximate faithfulness violations (small $\\epsilon$ perturbations to parameters) would address the most common theoretical concern.

Finally, even an informal comparison with one nonlinear ICA baseline (e.g., iVAE) on a shared synthetic benchmark would help contextualize the practical benefits of the nonparametric guarantee.

**Limitations:**

yes

**Strengths And Weaknesses:**

### Strengths

1. **Novel and general identifiability result**: This is, to my knowledge, the first fully nonparametric identifiability guarantee for task-relevant representations. The generality of the result (arbitrary mixing functions, disconnected sequences, interleaved tasks, arbitrary task ordering) is a meaningful theoretical advance.

2. **Clean two-level structure**: The decomposition into (a) task structure identification via CI testing and (b) representation identification via sparsity regularization is well-organized. Each level has a clear theorem, clear assumptions, and a clear algorithm. This modular structure makes both the contribution and its limitations easy to reason about.

3. **Collider-based insight is elegant**: Modeling each task as a collider in the SCM, so that task-relevant latents are parents of the task outcome, enables the use of conditional independence to distinguish relevant from irrelevant variables.

4. **Generalist superset property**: The result that generalist representations can only grow their support (never lose task-relevant variables) provides a clean theoretical justification for the common practice of fine-tuning generalist models for specialist tasks. This is a useful conceptual contribution beyond the identifiability results.

5. **Fully nonparametric setting**: The absence of parametric or structural constraints on the mixing function $f_t$ is a genuine strength.

### Weaknesses

1. **Standard assumptions deserve more discussion**: The theory relies on Markov and Faithfulness assumptions together with exact CI testing, which are standard in the causal discovery literature. However, the paper would benefit from a brief discussion of their practical limitations in this setting, particularly how faithfulness violations (e.g., near-cancellations in neural network parameterizations) and finite-sample CI errors in high-dimensional observation spaces might affect task structure recovery. A short sensitivity analysis on the CI test threshold would also be informative.

2. **Quantitative experiments only in the linear Gaussian setting**: The main quantitative experiment uses linear Gaussian SCMs, which is the most favorable setting for CI testing and does not stress-test the nonparametric claim of the theory. The SportsHHI and Flux experiments are qualitative demonstrations without ground-truth evaluation of representation quality. A nonlinear synthetic experiment with known ground truth would significantly strengthen the empirical contribution.

3. **No comparison with nonlinear ICA methods**: The paper does not compare representation quality against identifiable representation learning methods such as iVAE, SlowVAE, or CausalVAE, which target related problems under different assumptions. Even if the assumptions differ, an empirical comparison on shared benchmarks would help position the contribution and reveal practical trade-offs.

4. **Scalability of CI testing is unclear**: The method requires CI tests for all combinations of tasks $k$, time steps $v$, and latent variable indices $i$. With many tasks and long sequences, this becomes a large multiple-testing problem. The paper does not discuss correction for multiple comparisons or the computational cost of the CI testing phase in realistic settings.

---

> ### Author Rebuttal · Authors · 2026-03-31
>
> We sincerely appreciate the helpful feedback. In light of it, we add **new experiments, analyses, and discussions** as follows:
>
> ---
>
> **Q1.** More discussion on standard assumptions
>
> **A1.** Thanks for the insightful suggestions. We fully agree and have highlighted these, such as the violation of faithfulness:
>
> > “In finite-sample regimes, faithfulness can be sensitive to testing errors, and approximate violations may arise in practice, especially in high-dimensional settings where near-cancellations become more likely [1].”
>
> We further broaden the discussion in two directions.
>
> - Instead of faithfulness, one may consider alternative versions of Occam’s razor, such as sparsest Markov representation[2], frugality [3], SGS- or P- minimality [4, 5].
>
> - One may also explore weaker variants similar to adjacency-, orientation-, and triangle-faithfulness [6,7]. These relax the full faithfulness while still preserving the identifiability of key structural components.
>
> [1] Geometry of the Faithfulness Assumption
>
> [2] Learning Directed Acyclic Graph Models Based on Sparsest Permutations
>
> [3] The Frugal Inference of Causal Relations
>
> [4] Causation, Prediction, and Search
>
> [5] Causality: Models, Reasoning, and Inference
>
> [6] Adjacency-Faithfulness and Conservative Causal Inference
>
> [7] Detection of Unfaithfulness and Robust Causal Inference
>
>
> ---
>
> **Q2.** Sensitivity analysis on the test threshold.
>
> **A2.** Thanks for the suggestion. We vary it with $T=10$, $M=2$ and five runs:
>
> |Threshold|0.005|0.01|0.05|0.1|
> |-|-|-|-|-|
> |MCC|0.75±0.13|0.82±0.13|0.76±0.19|0.65±0.22|
> |Accuracy|0.81±0.05|0.85±0.10|0.79±0.09|0.70±0.18|
>
> Performance is stable across thresholds, with moderate values best; large thresholds introduce false positives.
>
> ---
>
> **Q3.** More nonlinear quantitative experiments.
>
> **A3.** Thanks for the suggestion. We add nonlinear experiments following Fig. 3, with MLPs for data generation and kernel-based CI tests. MCCs are reported below. Our method consistently outperforms baselines.
>
> |Method|T=8|T=10|T=12|T=14|T=16|T=18|T=20|
> |-|-|-|-|-|-|-|-|
> |CCA|0.25±0.13|0.17±0.11|0.24±0.08|0.27±0.10|0.24±0.15|0.31±0.08|0.26±0.10|
> |GroupLasso|0.44±0.13|0.28±0.13|0.35±0.06|0.32±0.11|0.28±0.11|0.19±0.15|0.21±0.17|
> |SelTask|0.42±0.16|0.46±0.18|0.30±0.12|0.32±0.16|0.25±0.10|0.41±0.17|0.44±0.16|
> |Ours|0.81±0.10|0.82±0.13|0.69±0.14|0.60±0.13|0.60±0.15|0.51±0.12|0.55±0.18|
>
> |Method|M=2|M=4|M=6|M=8|M=10|
> |-|-|-|-|-|-|
> |CCA|0.17±0.12|0.18±0.11|0.15±0.10|0.20±0.09|0.21±0.10|
> |GroupLasso|0.11±0.08|0.15±0.09|0.17±0.08|0.18±0.07|0.18±0.07|
> |SelTask|0.18±0.14|0.18±0.13|0.22±0.12|0.21±0.11|0.23±0.10|
> |Ours|0.35±0.13|0.40±0.16|0.42±0.15|0.41±0.14|0.43±0.13|
>
> **Simulation for Theorem 2:** Fig. 5 already uses nonlinear data (L400), and we further add new nonlinear baselines (A4).
>
> ---
>
> **Q4.** Comparison with nonlinear ICA methods.
>
> **A4.** Thank you for the suggestion. We add more discussion (e.g., SlowVAE), and compare with iVAE via the $R^2$ gap between relevant and irrelevant parts (larger is better, cf. Fig. 5) across different dimensionalities.
>
> |Method|10|20|30|40|50|60|70|80|90|100|
> |-|-|-|-|-|-|-|-|-|-|-|
> |iVAE|0.10±0.04|0.06±0.08|0.07±0.11|0.05±0.04|0.06±0.07|0.10±0.08|0.09±0.02|0.10±0.05|0.03±0.04|0.13±0.11|
> |Ours|0.63±0.07|0.66±0.06|0.65±0.04|0.66±0.05|0.72±0.08|0.66±0.07|0.65±0.04|0.63±0.09|0.70±0.09|0.65±0.07|
>
> We observe that iVAE fails to separate relevant from irrelevant components, while our method consistently achieves a large gap. This is expected, as iVAE relies on sufficient auxiliary information, which is not available in our setting.
>
> ---
>
> **Q5.** Scalability analysis of CI testing, both empirically and theoretically.
>
> **A5.** Thanks for the great question. We now conduct additional runtime analysis across seven datasets with four methods. To test the most computationally demanding case, we set $L=2$, vary $T\in[8,20]$, and set $M=T/5$. Results (seconds):
>
> |Method|8|10|12|14|16|18|20|
> |-|-|-|-|-|-|-|-|
> |CCA|0.013±0.009|0.013±0.000|0.016±0.000|0.018±0.000|0.024±0.000|0.027±0.000|0.035±0.000|
> |GroupLasso|11.220±3.527|26.769±5.488|34.846±4.153|36.304±3.110|58.315±0.872|59.728±1.453|82.287±0.154|
> |SelTask|0.692±0.040|1.123±0.002|1.400±0.114|1.334±0.117|1.823±0.041|2.114±0.009|2.398±0.011|
> |Ours|0.008±0.007|0.006±0.000|0.008±0.000|0.010±0.000|0.013±0.000|0.014±0.000|0.018±0.001|
>
> Ours is consistently faster, with runtime increasing smoothly in $T$.
>
> **Theoretical complexity**: Let $N=T/L$. Alg. 1 performs one CI test per segment pair per task, giving $O\left(M N^2\right)$ tests. Thus cost is linear in tasks $M$ and quadratic in segments $N$; latent dimension only affects CI cost.
>
> ---
>
> **Q6.** What happens when $L=1$?
>
> **A6.** Thank you for the question. The condition $L \ge 2$ ensures tasks are well-defined, as it provides the minimal granularity for tasks to exhibit temporal coherence. When $L=1$, a task degenerates into an instantaneous attribute, indistinguishable from observation or a noise spike at that step.

---

> > ### Author Rebuttal · Reviewer_Voe2 · 2026-04-03
> >
> > Thank you for your response.

---

> > > ### Author Response · Authors · 2026-04-03
> > >
> > > Thank you very much for your support and insightful feedback!

---

### Decision · Program_Chairs · 2026-04-30

**Decision:**

Accept (regular)

**Comment:**

The paper studies the learning of task-relevant latent representations from observations, in a nonparametric fashion, without interventions or structural constraints. Two steps are performed: learning temporal task structure and then learning task-relevant representation within each time step. Theoretical results as well as synthetic and real-world experimental evidence are provided.

Opinions from reviewers were mixed, and the majority of reviewers argued for acceptance. I finally decided for acceptance. For a camera-ready version, please take into account the comments from all reviewers regarding for instance discussion of assumptions, clarity in the theoretical exposition and intuitive visualizations.